# Loss of Ptpmt1 limits mitochondrial utilization of carbohydrates and leads to muscle atrophy and heart failure in tissue-specific knockout mice

Hong Zheng[1,2†], Qianjin Li[1†], Shanhu Li[1,2†], Zhiguo Li[1], Marco Brotto[3], Daiana Weiss[4], Domenick Prosdocimo[5], Chunhui Xu[1], Ashruth Reddy[1], Michelle Puchowicz[6], Xinyang Zhao[7], M Neale Weitzmann[4], Mukesh K Jain[5], Cheng-Kui Qu[1,2*]

[1]Department of Pediatrics, Children Healthcare of Atlanta, Emory University School of Medicine, Atlanta, United States; [2]Department of Medicine, Case Western Reserve University, Cleveland, United States; [3]College of Nursing & Health Innovation, University of Texas-Arlington, Arlington, United States; [4]Department of Medicine, Emory University School of Medicine, Atlanta, United States; [5]Case Cardiovascular Research Institute, Department of Medicine, Case Western Reserve University, Cleveland, United States; [6]Case Mouse Metabolic Phenotyping Center, Case Western Reserve University, Cleveland, United States; [7]Department of Biochemistry and Molecular Genetics, University of Alabama at Birmingham, Birmingham, United States

*For correspondence:
cheng-kui.qu@emory.edu

†These authors contributed equally to this work

**Abstract** While mitochondria in different tissues have distinct preferences for energy sources, they are flexible in utilizing competing substrates for metabolism according to physiological and nutritional circumstances. However, the regulatory mechanisms and significance of metabolic flexibility are not completely understood. Here, we report that the deletion of *Ptpmt1*, a mitochondria-based phosphatase, critically alters mitochondrial fuel selection – the utilization of pyruvate, a key mitochondrial substrate derived from glucose (the major simple carbohydrate), is inhibited, whereas the fatty acid utilization is enhanced. *Ptpmt1* knockout does not impact the development of the skeletal muscle or heart. However, the metabolic inflexibility ultimately leads to muscular atrophy, heart failure, and sudden death. Mechanistic analyses reveal that the prolonged substrate shift from carbohydrates to lipids causes oxidative stress and mitochondrial destruction, which in turn results in marked accumulation of lipids and profound damage in the knockout muscle cells and cardiomyocytes. Interestingly, *Ptpmt1* deletion from the liver or adipose tissue does not generate any local or systemic defects. These findings suggest that Ptpmt1 plays an important role in maintaining mitochondrial flexibility and that their balanced utilization of carbohydrates and lipids is essential for both the skeletal muscle and the heart despite the two tissues having different preferred energy sources.

## eLife assessment

This paper provides a **useful** set of data examining the role of PTPMT1, a mitochondria-based phosphatase, in mitochondrial fuel selection. The data were collected and analyzed using **solid** methodology and can be used as a starting point for further studies that build on the findings here.

**eLife digest** Cells are powered by mitochondria, a group of organelles that produce chemical energy in the form of molecules called ATP. This energy is derived from the breakdown of carbohydrates, fats, and proteins.

The number of mitochondria in a cell and the energy source they use to produce ATP varies depending on the type of cell. Mitochondria can also switch the molecules they use to produce energy when the cell is responding to stress or disease.

The heart and the skeletal muscles – which allow movement – are two tissues that require large amounts of energy, but it remained unknown whether disrupting mitochondrial fuel selection affects how these tissues work.

To answer these questions, Zheng, Li, Li et al. investigated the role of an enzyme found in mitochondria called Ptpmt1. Genetically deleting Ptpmt1 in the heart and skeletal muscle of mice showed that while the development of these organs was not affected, mitochondria in these cells switched from using carbohydrates to using fats as an energy source. Over time, this shift damaged both the mitochondria and the tissues, leading to muscle wasting, heart failure, and sudden death in the mice. This suggests that balanced use of carbohydrates and fats is essential for the muscles and heart.

These findings imply that long-term use of medications that alter the fuel that mitochondria use may be detrimental to patients' health and could cause heart dysfunction. This may be important for future drug development, as well as informing decisions about medication taken in the clinic.

## Introduction

Mitochondria, the powerhouse of the cell, produce energy in the form of ATP from the breakdown of carbohydrates, fats, and proteins. Mitochondrial abundance and their preferences for metabolic substrates differ in various cell types (*Smith et al., 2018*). While mitochondria are abundant in red skeletal muscles such as Soleus, and the heart, the mitochondrial content in white muscles, such as Extensor Digitorum Longus (EDL), is much lower. Mitochondria in the skeletal muscle and heart have distinct preferences for energy sources – skeletal muscle mitochondria prefer glucose (the major simple carbohydrate) as the substrate, whereas cardiac mitochondria mainly use fatty acids as the fuel. Moreover, different types of skeletal muscle cells (fibers) utilize glucose to produce energy in different mechanisms – in slow-twitch oxidative muscle fibers glucose (via pyruvate, a critical metabolite derived from glucose) is predominantly oxidized within the mitochondria to generate energy, whereas in fast-twitch muscle fibers pyruvate is primarily reduced to lactate in the cytosol promoting glycolytic flux to rapidly produce energy and support quick contractions. Nevertheless, both muscle and heart mitochondria are flexible in selecting substrates in response to stress (exercise, fasting, etc.) and under disease conditions (heart failure, diabetes, etc.) (*Bertero and Maack, 2018*; *Brown et al., 2017*; *Nabben et al., 2018*; *Schulze et al., 2016*). It remains to be determined how mitochondrial utilization of various competing metabolic substrates is coordinated and whether disruption of mitochondrial flexibility in fuel selection affects the function of the heart and skeletal muscle.

Ptpmt1, encoded by nuclear DNA, is localized to the mitochondrion and anchored at the inner membrane (*Pagliarini et al., 2005*). It dephosphorylates phosphatidylinositol phosphates (PIPs). PIPs are a class of membrane phospholipids that bind to a distinctive set of effector proteins, thereby regulating a characteristic suite of cellular processes, including membrane trafficking and ion channel/transporter functions (*Balla, 2006*; *Gamper and Shapiro, 2007*). Ptpmt1 is also involved in the synthesis of cardiolipin by converting phosphotidylglecerol phosphate to phosphotidylglecerol, the precursor of cardiolipin (*Zhang et al., 2011*). Global knockout of *Ptpmt1* results in developmental arrest and post-implantation lethality (*Zhang et al., 2011*; *Shen et al., 2011*). Our previous studies suggest that Ptpmt1 facilitates mitochondrial metabolism largely by dephosphorylation of downstream PIP substrates (*Yu et al., 2013*; *Shen et al., 2009*) that appear to inhibit mitochondrial oxidative phosphorylation but enhance cytosolic glycolysis by activation of mitochondrial uncoupling protein 2 (Ucp2) (*Yu et al., 2013*), a transporter of four-carbon (C4) dicarboxylate intermediates of the tricarboxylic acid (TCA) cycle that has been shown to regulate cellular energetics by limiting mitochondrial oxidation of glucose (*Vozza et al., 2014*; *Bouillaud, 2009*; *Diano and Horvath, 2012*; *Pecqueur et al., 2008*; *Samudio et al., 2009*). *Ptpmt1* is highly expressed in the heart and skeletal muscle (*Shen et al.,*

*2011*). In the present study, we exploit *Ptpmt1* tissue-specific knockout models to address the role and mechanisms of Ptpmt1-facilitated mitochondrial metabolism in these tissues. We find that Ptpmt1 plays an important role in facilitating mitochondrial utilization of carbohydrates and that balanced fuel selection is essential for maintaining both muscle and heart functions despite the two tissues having distinct preferences for energy sources.

## Results

### Knockout of *Ptpmt1* from skeletal muscles results in defective contractile function and progressive muscle atrophy

To determine the role of Ptpmt1-mediated metabolism in the skeletal muscle and heart, we generated tissue-specific *Ptpmt1* knockout mice (*Ptpmt1^{fl/fl}/Ckm-Cre^+*) by crossing *Ptpmt1 floxed* mice (*Yu et al., 2013*) and muscle creatine kinase promoter-driven *Cre* transgenic mice (*Ckm-Cre*), which express the Cre recombinase in the skeletal muscle and heart starting at embryonic day 13 (*Brüning et al., 1998*). Quantitative reverse transcription PCR showed ~95% and ~80% deletion of *Ptpmt1* in the skeletal muscle and heart in *Ptpmt1^{fl/fl}/Ckm-Cre^+* mice, respectively (*Figure 1A*). These animals were indistinguishable from their wild-type (WT, *Ptpmt1^{+/+}/Ckm-Cre^+*) littermates up to 3 months of age, suggesting that *Ptpmt1* deletion did not affect muscle or heart development. The knockout mice exhibited reduced weight gain starting at 4–5 months and their body sizes were significantly smaller than those of control mice at 8 months (*Figure 1A*). Muscle wasting was observed in *Ptpmt1^{fl/fl}/Ckm-Cre^+* mice at 8 months or older (*Figure 1A*). Histopathological examination of skeletal muscles of these knockout mice revealed characteristic changes of muscle atrophy – marked variations in muscle fiber diameters and the presence of ghost fibers or more interstitial space (*Figure 1B*). In addition, Masson's Trichrome staining revealed increased collagen-rich fibrotic regions in *Ptpmt1* knockout muscles (*Figure 1B*). We then examined *Ptpmt1* knockout mice at 6–8 months (prior to the onset of muscle wasting). *Ptpmt1^{fl/fl}/Ckm-Cre^+* mice became fatigued faster and fell more quickly than *Ptpmt1^{+/+}/Ckm-Cre^+* control animals in wire hang tests, indicating muscle weakness (*Figure 1C*). In treadmill exercise tests, both male and female knockout mice displayed reduced maximum speed (*Figure 1D*), endurance (*Figure 1E*), and distance of the run (*Figure 1—figure supplement 1A*). To determine whether the reduced muscle strengths of the knockout mice resulted from muscle cell-intrinsic defects, isometric contractile properties of isolated muscles were examined. Ex vivo force–frequency measurements were performed on Soleus, a more oxidative muscle with a higher percentage of slow-twitch Type I fibers, and EDL, a more glycolytic muscle with a higher percentage of fast-twitch Type II fibers. Both knockout Soleus and EDL showed decreased optimal length, the length of the muscle at which maximal force was achieved (*Figure 1—figure supplement 1B, C*). Specific force production of *Ptpmt1* knockout Soleus (*Figure 1—figure supplement 1D*) and EDL (*Figure 1—figure supplement 1E*) was decreased. However, normalized force production was decreased only in knockout Soleus (*Figure 1F*), but not in knockout EDL (*Figure 1G*). These data suggest that Ptpmt1 plays a more important role in oxidative than glycolytic muscle fibers although both knockout Soleus and EDL developed muscle atrophy subsequently.

To determine the impact of Ptpmt1 depletion from the skeletal muscle and heart on the metabolism of the whole body, we measured plasma glucose and fatty acid levels and found that they were comparable in *Ptpmt1^{fl/fl}/Ckm-Cre^+* and *Ptpmt1^{+/+}/Ckm-Cre^+* mice (*Figure 1—figure supplement 2A*). Plasma levels of lactate (*Figure 1—figure supplement 2B*) and lipid metabolic products (*Figure 1—figure supplement 2C*) in *Ptpmt1* knockout mice were also relatively normal (except that cholesterol and nonesterified fatty acids levels were decreased and marginally increased, respectively, in female knockout mice). Adipokine array assays of the plasma showed no or subtle differences in Adiponectin, Leptin, Lipocalin-2, Angptl3, Resistin, Fgf-21, Dppiv, Fetuin A, Igf-1, etc. between *Ptpmt1* knockout and control mice (*Figure 1—figure supplement 2D*). In addition, no differences in glucose tolerance tests were observed between *Ptpmt1* knockout and control mice (*Figure 1—figure supplement 2E*). Furthermore, we assessed the expression levels of key enzymes involved in glucose and lipid metabolism in *Ptpmt1* knockout muscles. No changes in Hk2, Pkm1, Ldha, Dicer1, Mcad (Acadm), Acadl, Hadha, Cpt2, or Fabp3 were found. However, the expression of Cpt1B was increased in *Ptpmt1* knockout muscles (*Figure 1—figure supplement 2F, G*). These results suggest that systemic glucose

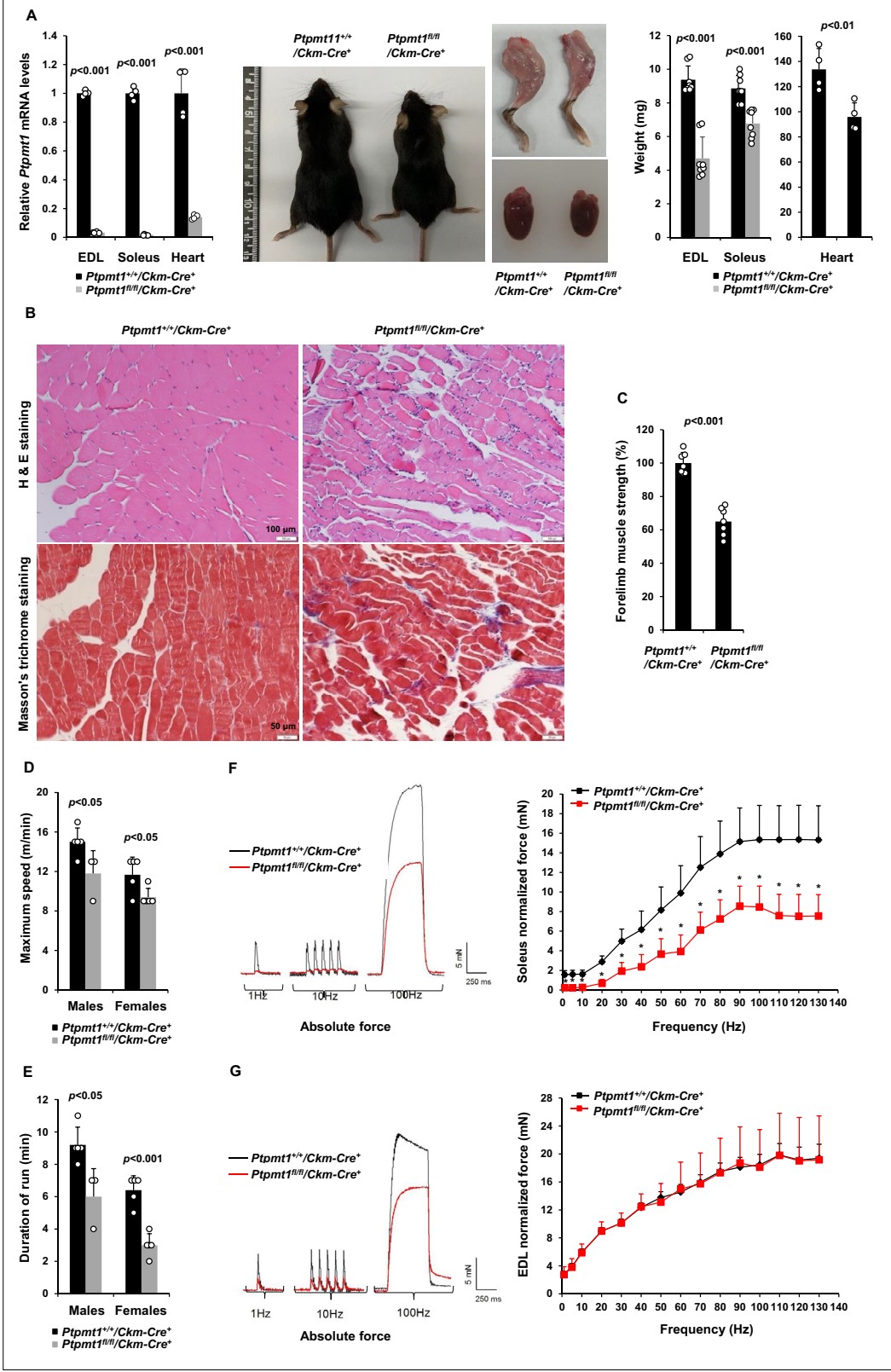

**Figure 1.** Deletion of *Ptpmt1* from skeletal muscles results in defective contractility and progressive muscle atrophy. (**A**) Representative 8-month-old of *Ptpmt1^{fl/fl}/Ckm-Cre^+* and *Ptpmt1^{+/+}/Ckm-Cre^+* mice, and hind limbs and hearts dissected from these mice were photographed. Extensor Digitorum Longus (EDL) and Soleus (*n* = 8 mice/genotype), and heart (*n* = 4 mice/genotype) weights were measured. *Ptpmt1* mRNA levels in EDL, Soleus,

*Figure 1 continued on next page*

*Figure 1 continued*

and heart tissues were determined by quantitative reverse transcription PCR (qRT-PCR) (*n* = 4 mice/genotype). (**B**) Skeletal muscle sections prepared from 8-month-old *Ptpmt1*[+/+]/*Ckm-Cre*[+] and *Ptpmt1*[fl/fl]/*Ckm-Cre*[+] mice were processed for Hematoxylin and Eosin (H&E) staining and Masson's Trichrome staining. One representative image from 3 mice/genotype is shown. (**C**) Six-month-old *Ptpmt1*[+/+]/*Ckm-Cre*[+] and *Ptpmt1*[fl/fl]/*Ckm-Cre*[+] mice (*n* = 7/genotype) were subjected to wire hang tests. Relative forelimb muscle strength was determined. (**D, E**) Seven- to eight-month-old *Ptpmt1*[+/+]/*Ckm-Cre*[+] (*n* = 5 males and 5 females) and *Ptpmt1*[fl/fl]/*Ckm-Cre*[+] (*n* = 3 males and 5 females) mice were assessed by treadmill exercise tests as described in Materials and methods. Maximum speed (**D**) and duration of the run (**E**) were recorded. Soleus (**F**) and EDL (**G**) dissected from 7- to 8-month-old *Ptpmt1*[+/+]/*Ckm-Cre*[+] and *Ptpmt1*[fl/fl]/*Ckm-Cre*[+] mice (*n* = 3 mice, 6 muscles/genotype) were subjected to ex vivo isometric force measurements. Specific contractile forces produced at the indicated frequencies of stimulation were normalized to the physiological cross-sectional area. Shown on the left are representative absolute forces produced at 1, 10, and 100 Hz. *p <0.05.

The online version of this article includes the following source data and figure supplement(s) for figure 1:

**Figure supplement 1.** Specific force production is decreased in Soleus and Extensor Digitorum Longus (EDL) isolated from *Ptpmt1*[fl/fl]/*Ckm-Cre*[+] mice.

**Figure supplement 2.** Plasma levels of glucose and lipids are comparable in *Ptpmt1*[fl/fl]/*Ckm-Cre*[+] mice and *Ptpmt1*[+/+]/*Ckm-Cre*[+] littermates.

**Figure supplement 2—source data 1.** Uncropped immunoblotting images of *Figure 1—figure supplement 2G*.

and lipid metabolism in these tissue-specific knockout mice was not significantly changed, but fatty acid partitioning for oxidation in the mitochondria was elevated in *Ptpmt1* knockout muscles.

## Ptpmt1 depletion ultimately leads to mitochondrial damage and bioenergetic stress in knockout skeletal muscles

Gömöri trichrome staining revealed ragged red fibers in *Ptpmt1* knockout skeletal muscles (*Figure 2A*), as observed in various types of human mitochondrial myopathies. Electron microscopic analyses showed that interfibrillar mitochondria in knockout Soleus and EDL were disorganized and enlarged, in contrast to normal mitochondria that typically reside in pairs and position on either side of the Z-disc in control muscles. In addition, there was a massive accumulation of subsarcolemmal mitochondria in the knockout muscles, whereas the content of intermyofibrillar mitochondria decreased (*Figure 2B*). Although EDL, unlike Soleus, uses more cytosolic glycolysis than mitochondrial oxidative phosphorylation for energy production, the mitochondrial structural changes appeared to be more severe in EDL than Soleus in the knockout mice (*Figure 2B*). Furthermore, a massive accumulation of intramyofibrillar lipids were detected by Oil Red O staining in 6 months or older *Ptpmt1* knockout muscles (*Figure 2C*).

Total mitochondrial content in *Ptpmt1* knockout Soleus and EDL also slightly decreased compared to that in control muscles (*Figure 2D*). ATP levels in both knockout Soleus and EDL decreased significantly compared to WT counterparts (*Figure 2E*). Consistent with these data, the energetic stress sensor, AMP-activated kinase (Ampk), was highly activated in *Ptpmt1*[fl/fl]/*Ckm-Cre*[+] muscles as determined by the phosphorylation levels of Thr[172] (*Figure 2F*). Acetyl-CoA carboxylase (Acc), a negative regulator of fatty acid oxidation, was inhibited, as evidenced by a marked increase in the inhibitory phosphorylation of this enzyme (*Figure 2F*), implying increased fatty acid oxidation in Ptpmt1-depleted mitochondria. In agreement with previous findings that Ampk negatively regulates mTor signaling (*Hardie, 2015*; *Liang and Mills, 2013*), mTor was substantially inhibited. Activities of S6k, S6, and 4E-bp1, key downstream components of mTor signaling, were also concomitantly decreased (*Figure 2F*), indicating diminished anabolic activities and consistent with the muscle atrophy phenotype. Interestingly, Akt activities, as reflected by its phosphorylation levels (Ser[473] and Thr[308]), were increased in *Ptpmt1* knockout skeletal muscles, recapitulating mTor and raptor (a component of the mTor complex 1) knockout muscles (*Bentzinger et al., 2008*; *Risson et al., 2009*). Likely due to the cross-talk between Akt and Ras signaling pathways, Erk activities were also slightly increased in these knockout muscles (*Figure 2F*).

Given that muscle atrophy developed in *Ptpmt1*[fl/fl]/*Ckm-Cre*[+] mice after 6–8 months, to assess the direct impact of *Ptpmt1* deletion on mitochondrial metabolism, we examined mitochondria in young *Ptpmt1* knockout mice before the tissue damage. Expression levels of electron transport chain

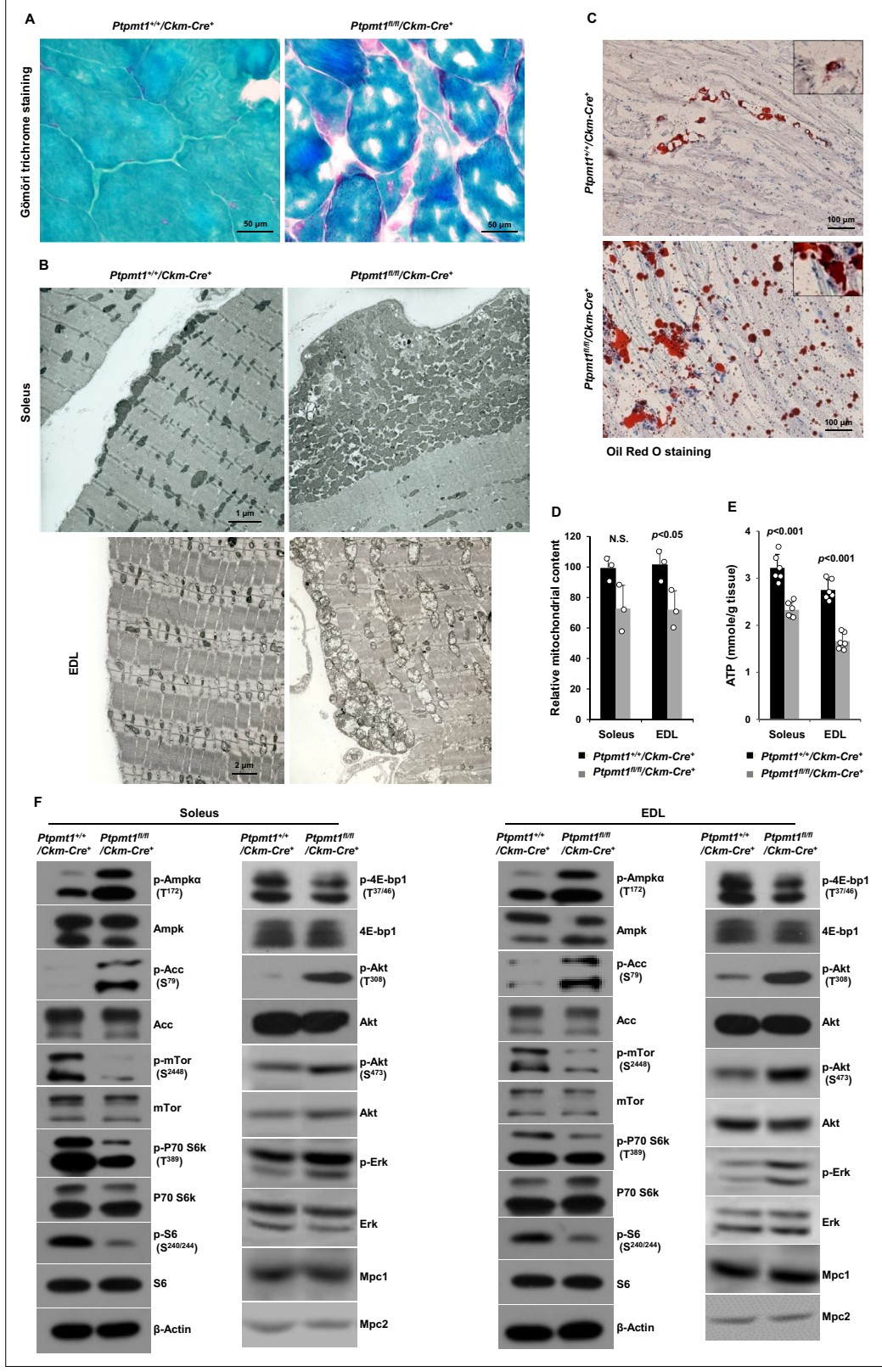

**Figure 2.** Ptpmt1 loss ultimately leads to abnormal mitochondrial distribution, structural damage, and bioenergetic stress in skeletal muscles. (**A**) Skeletal muscle sections prepared from 6-month-old *Ptpmt1⁺/⁺/Ckm-Cre⁺* and *Ptpmt1ᶠˡ/ᶠˡ/Ckm-Cre⁺* mice were processed for *Gömöri* trichrome staining. One representative image from 3 mice/genotype is shown. (**B**) Soleus and Extensor Digitorum Longus (EDL) dissected from 8-month-old *Ptpmt1ᶠˡ/*

*Figure 2 continued on next page*

*Figure 2 continued*

*Ckm-Cre+* and *Ptpmt1+/+/Ckm-Cre+* mice were processed for transmission electron microscopic examination. One representative image from 3 mice/genotype is shown. (**C**) Skeletal muscle sections prepared from 6-month-old *Ptpmt1+/+/Ckm-Cre+* and *Ptpmt1fl/fl/Ckm-Cre+* mice were processed for Oil Red O staining to visualize lipids. One representative picture from 3 mice/genotype is shown. (**D**) Total DNA was extracted from Soleus and EDL dissected from 8-month-old *Ptpmt1fl/fl/Ckm-Cre+* and *Ptpmt1+/+/Ckm-Cre+* mice (*n* = 3/genotype). Mitochondrial content was estimated by comparing the mitochondrial gene cytochrome B DNA levels to the nuclear gene 18S DNA levels by qPCR. (**E**) Total ATP levels in Soleus and EDL dissected from 8-month-old *Ptpmt1fl/fl/Ckm-Cre+* and *Ptpmt1+/+/Ckm-Cre+* mice (*n* = 6/genotype) were determined. (**F**) Whole cell lysates prepared from the Soleus and EDL isolated from 7-month-old *Ptpmt1fl/fl/Ckm-Cre+* and their control mice were examined by immunoblotting with the indicated antibodies. Representative results from 3 mice/genotype are shown.

The online version of this article includes the following source data and figure supplement(s) for figure 2:

**Source data 1.** Uncropped immunoblotting images of *Figure 2F*.

**Figure supplement 1.** Energy homeostasis is initially maintained in young *Ptpmt1fl/fl/Ckm-Cre+* skeletal muscles.

**Figure supplement 1—source data 1.** Uncropped immunoblotting images of *Figure 2—figure supplement 1C, D*.

---

complexes in these Ptpmt1-ablated mitochondria were not changed (*Figure 2—figure supplement 1A*). Total cellular ATP levels in *Ptpmt1* knockout muscles at this age were comparable to those in control tissues (*Figure 2—figure supplement 1B*), and the knockout muscles did not show bioenergetic/metabolic stress (*Figure 2—figure supplement 1C*). We also checked Lc3-I/II levels in *Ptpmt1* knockout muscles and found no evidence of elevated autophagic activities (*Figure 2—figure supplement 1D*). These results indicate that the decrease in total cellular ATP levels in older knockout muscles (*Figure 2E*) was a secondary not direct effect of Ptpmt1 depletion.

## Ptpmt1 loss impairs mitochondrial utilization of pyruvate, whereas the fatty acid utilization is enhanced

We next sought to address the mechanisms by which Ptpmt1 loss impacts mitochondrial metabolism. To avoid secondary effects, we examined mitochondrial metabolic activities of *Ptpmt1* knockout muscles in 3-month-old *Ptpmt1fl/fl/Ckm-Cre+* and *Ptpmt1+/+/Ckm-Cre+* mice. Knockout muscle tissues showed decreased maximal reserve oxidative capacities in the presence of full metabolic substrates (*Figure 3A*), consistent with our previous observations from other *Ptpmt1* knockout cell types (*Yu et al., 2013*; *Shen et al., 2009*; *Zheng et al., 2018*). In addition, we isolated mitochondria from young skeletal muscles and assessed mitochondrial metabolism in the presence of a single metabolic substrate by real-time measurement of ATP synthesis-driven oxygen consumption. When pyruvate, a critical metabolite derived from glucose, was provided as the sole substrate, ATP synthesis-driven oxygen consumption in Ptpmt1-deficient mitochondria was markedly decreased (*Figure 3B*), suggesting that mitochondrial utilization of pyruvate was diminished in the absence of Ptpmt1. Interestingly, we observed enhanced oxygen consumption in these mitochondria when fatty acids were supplied as the fuel (*Figure 3C*). A slightly increased respiratory response in Ptpmt1-depleted mitochondria was also detected when glutamate, another mitochondrial substrate, was provided (*Figure 3D*). Importantly, feeding with succinate, the substrate for Complex II of the electron transport chain, resulted in similar oxygen consumption in Ptpmt1 null and control mitochondria (*Figure 3E*), validating the intact function of the electron transport chain in the knockout mitochondria and excluding flavin adenine dinucleotide and quinone-linked deficiencies. These mechanistic data suggest that Ptpmt1 plays an important role in facilitating carbohydrate oxidation in mitochondria.

Consistent with the above notion, intramitochondrial pyruvate levels in the mitochondria freshly isolated from *Ptpmt1* knockout muscles decreased by half (*Figure 3F*). Levels of α-ketoglutarate (α-KG), an important metabolite in the TCA cycle, also decreased in the knockout mitochondria (*Figure 3G*), although steady-state mitochondrial acetyl-CoA levels were not changed (*Figure 3H*). To further determine if pyruvate transport efficiency may be reduced in Ptpmt1-ablated mitochondria, we incubated fresh mitochondria with pyruvate and measured acute production of α-KG from extramitochondrial pyruvate. As shown in *Figure 3I*, α-KG production in Ptpmt1-deficient mitochondria was indeed decreased compared to that in control mitochondria. Notably, the activity of pyruvate dehydrogenase (Pdh), the enzyme that converts pyruvate to acetyl-CoA to feed the TCA cycle, was not affected in the

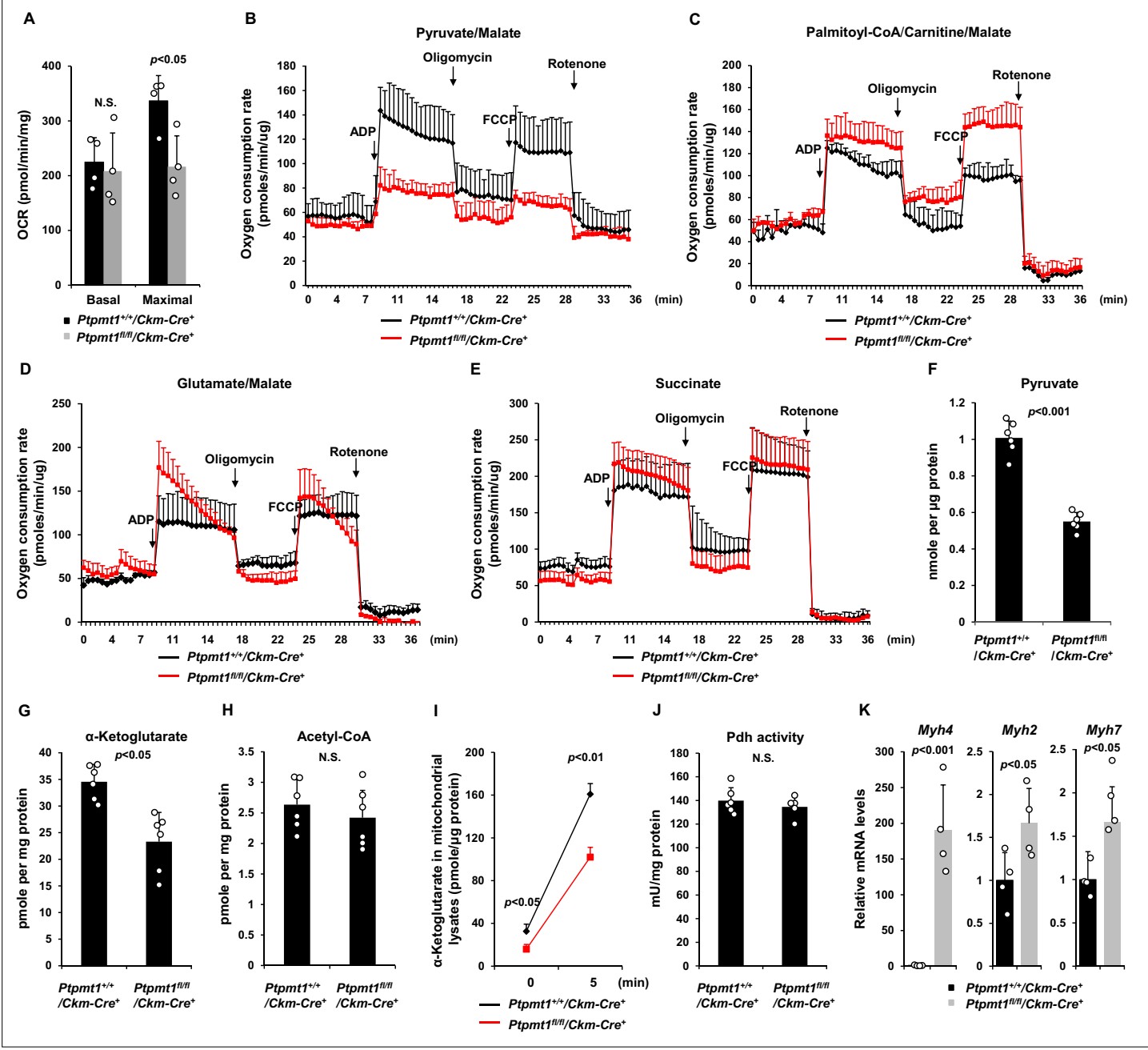

**Figure 3.** Ptpmt1 ablation impairs mitochondrial utilization of pyruvate, whereas the fatty acid utilization is enhanced. (**A**) Muscle cross-sections prepared with biopsy punches from *Ptpmt1fl/fl/Ckm-Cre+* and *Ptpmt1+/+/Ckm-Cre+* mice (n = 4/genotype) at 4 months of age were measured for oxygen consumption rates (OCRs) at the basal level and following the addition of oligomycin (8 µM), FCCP (4 µM), and antimycin A/rotenone (1 µM). (**B–E**) Mitochondria were isolated from the skeletal muscles dissected from *Ptpmt1fl/fl/Ckm-Cre+* and *Ptpmt1+/+/Ckm-Cre+* mice (n = 3/genotype) at 3 months of age. Mitochondrial oxygen consumption (10 µg of mitochondrial protein) was measured in the presence of pyruvate (5 mM)/malate (5 mM) (**B**), palmitoyl-CoA (40 µM)/carnitine (40 µM)/malate (5 mM) (**C**), glutamate (5 mM)/malate (5 mM) (**D**), or succinate (10 mM) (**E**), following the addition of ADP (4 mM), oligomycin (1.5 µM), FCCP (4 µM), and antimycin A/rotenone (1 µM). Experiments were repeated three times with three independent pairs of mice. Similar results were obtained in each experiment. Levels of pyruvate (**F**), α-ketoglutarate (α-KG) (**G**), and acetyl-CoA (**H**) in the lysates of the mitochondria isolated from the above skeletal muscles were measured (n = 6/genotype). (**I**) Mitochondria freshly isolated from the skeletal muscles of *Ptpmt1fl/fl/Ckm-Cre+* and *Ptpmt1+/+/Ckm-Cre+* mice (n = 4/genotype) were washed three times in Mitochondrial Assay Solution (MAS) buffer. The mitochondria were then incubated with pyruvate (5 mM)/malate (5 mM) and ADP (4 mM) at 37°C. Five min later, mitochondria were collected, washed, and lysed. α-KG levels in the mitochondrial lysates were measured. (**J**) Pyruvate dehydrogenase (Pdh) activities in the mitochondrial lysates were determined (n = 5–6 mice/genotype). (**K**) *Myh4*, *Myh2*, and *Myh7* mRNA levels in the skeletal muscles dissected from 3-month-old *Ptpmt1+/+/Ckm-Cre+* and *Ptpmt1fl/fl/Ckm-Cre+* mice (n = 4/genotype) were determined by quantitative reverse transcription PCR (qRT-PCR).

knockout mitochondria (*Figure 3J*). These observations together with that the mitochondrial pyruvate carrier/transporter (Mpc) was comparably expressed in Ptpmt1-ablated mitochondria (*Figure 2F*) suggest that Ptpmt1 null mitochondria were impaired in uptaking pyruvate. Possibly due to an adaptive response to the inhibition of pyruvate oxidation in the mitochondria, skeletal muscle fiber-type switching occurred in *Ptpmt1^{fl/fl}/Ckm-Cre^+* mice. Muscle fibers in these knockout mice switched from oxidative to glycolytic fibers, as evidenced by a dramatic increase (~170-fold) in the levels of *Myh4*, a marker of glycolytic fast-twitch Type 2B fibers in *Ptpmt1* knockout mice (*Figure 3K*).

## *Ptpmt1^{fl/fl}/Ckm-Cre^+* mice manifest late-onset cardiac dysfunction

The heart in *Ptpmt1^{fl/fl}/Ckm-Cre^+* mice was less impacted than the skeletal muscle although *Ptpmt1* was ~80% deleted from the heart (*Figure 1A*). At 7 months, when skeletal muscles in these mice displayed atrophy and myocyte damage, no evident histological changes were observed in heart tissues (*Figure 4—figure supplement 1A*). Echocardiographic examination showed no difference in left ventricle (LV) contractility in *Ptpmt1* knockout mice as reflected by similar LV fractional shortening (FS), ejection fraction (EF), and the ratio of peak velocity of early to late filling of mitral inflow (E/A) between knockout and control mice (*Figure 4—figure supplement 1B*). Mitochondrial content in *Ptpmt1^{fl/fl}/Ckm-Cre^+* heart tissues was comparable to that in *Ptpmt1^{+/+}/Ckm-Cre^+* control tissues (*Figure 4—figure supplement 1C*). Total ATP levels in *Ptpmt1* knockout heart tissues were marginally but not significantly decreased (*Figure 4—figure supplement 1D*). Moreover, mitochondria in knockout cardiomyocytes showed minimal structural changes (*Figure 4—figure supplement 1E*). Consistent with these observations, no bioenergetic/metabolic stress was detected in the knockout cardiomyocytes according to the activation status of Ampk and mTor signaling (*Figure 4—figure supplement 1F*).

However, severe cardiac dysfunction was observed in 10- to 12-month-old *Ptpmt1^{fl/fl}/Ckm-Cre^+* mice. Histopathological examination revealed that cardiomyocytes in these animals had uneven sizes and were disorganized (*Figure 4A*). In addition, fibrotic lesions, indicative of myocardial damage, were observed. Echocardiographic examination showed that LV contractility, as measured by FS and EF in M-mode echocardiographic tracing of LV (*Figure 4B-D*) and recorded by M-mode echocardiography movies (*Figure 4—video 1* and *Figure 4—video 2*), was markedly decreased in *Ptpmt1* knockout mice. The E/A ratio determined in pulsed-wave Doppler recording of mitral valve inflow was doubled in the *Ptpmt1^{fl/fl}/Ckm-Cre^+* heart (*Figure 4E, F*). Collectively, these data suggest that Ptpmt1 depletion also impaired cardiac function albeit later than skeletal muscle dysfunction.

## Heart-specific deletion of *Ptpmt1* leads to dilated cardiomyopathy and heart failure

Given that *Ptpmt1* was only ~80% deleted from the heart in *Ptpmt1^{fl/fl}/Ckm-Cre^+* mice and that impaired skeletal muscle function in these animals may potentially complicate the heart phenotypes, to further investigate the role of Ptpmt1-mediated metabolism specifically in cardiac muscles, heart-specific *Ptpmt1* knockout mice (*Ptpmt1^{fl/fl}/Myh6-Cre^+*) were generated by crossing *Ptpmt1* conditional mice (*Yu et al., 2013*) and *Myh6-Cre* transgenic mice, which express the Cre recombinase specifically in cardiomyocytes (*Agah et al., 1997*). These knockout mice appeared healthy in the first several months although *Ptpmt1* was nearly completely deleted from the heart (but not in Soleus and EDL) (*Figure 5—figure supplement 1A*). No obvious tissue morphological changes or fibrotic lesions were found in *Ptpmt1* knockout hearts (*Figure 5—figure supplement 1B*), confirming that Ptpmt1-mediated metabolism is dispensable for the development of the heart. Echocardiography also showed normal heart function in 3-month-old *Ptpmt1^{fl/fl}/Myh6-Cre^+* mice, as evidenced by normal FS, EF, and E/A ratios (*Figure 5—figure supplement 1C–E*). Knockout mice did not exhibit any defects during treadmill exercises at 6 months of age (*Figure 5—figure supplement 1F*). However, all *Ptpmt1^{fl/fl}Myh6-Cre^+* knockout mice died at 10–16 months, and they often succumbed suddenly (*Figure 5A*) although their body weights were relatively normal. *Ptpmt1* knockout hearts showed enlarged ventricular chambers. Severe structural damage in cardiac myocytes and tremendous fibrotic lesions were observed in the knockout hearts (*Figure 5B*). Immunostaining of cleaved caspase 3 illustrated increased apoptosis in *Ptpmt1^{fl/fl}/Myh6-Cre^+* cardiomyocytes as compared to control cells (*Figure 5C*). Echocardiography revealed a profound dilated cardiomyopathy and heart failure in *Ptpmt1^{fl/fl}/Myh6-Cre^+* mice, including both systolic and diastolic LV dilation, thinning of ventricular walls, depression of EF and FS,

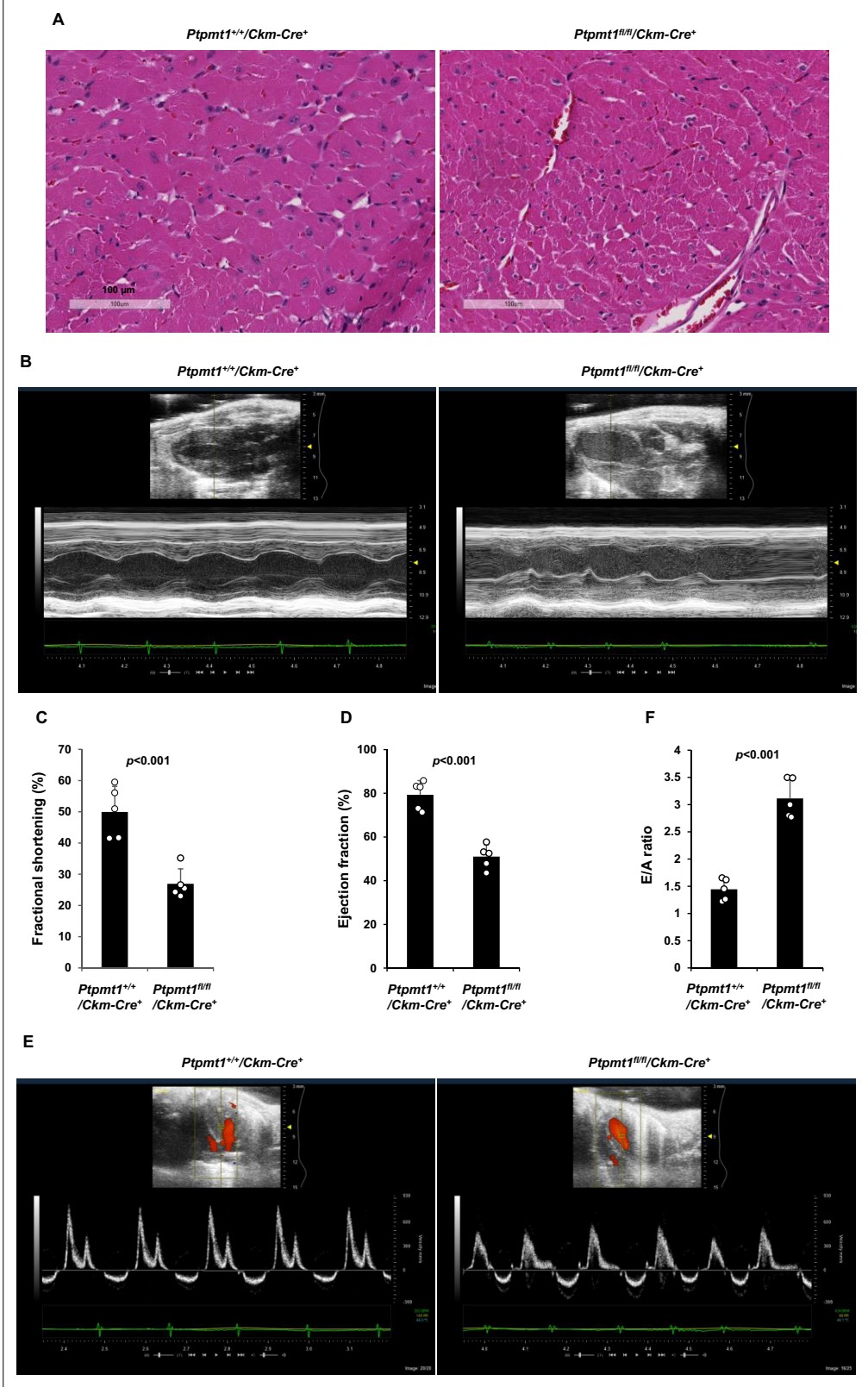

**Figure 4.** *Ptpmt1^{fl/fl}/Ckm-Cre^+* mice manifest late-onset cardiac dysfunction. (**A**) Heart tissue sections prepared from 12-month-old *Ptpmt1^{+/+}/Ckm-Cre^+* and *Ptpmt1^{fl/fl}/Ckm-Cre^+* mice were processed for Hematoxylin and Eosin (H&E) staining. One representative image from 3 mice/genotype is shown. (**B–F**) Cardiac morphology and function of 12-month-old *Ptpmt1^{+/+}/Ckm-Cre^+* and *Ptpmt1^{fl/fl}/Ckm-Cre^+* mice (*n* = 5/genotype) were examined

*Figure 4 continued on next page*

*Figure 4 continued*

by echocardiography. Representative long-axis views in M-mode echocardiographic tracing of left ventricle (LV) are shown (**B**). LV fractional shortening (FS) (**C**) and LV ejection fraction (EF) (**D**) were determined. Representative pulsed-wave Doppler recordings of mitral valve inflow are shown (**E**). Ratios of peak velocity of early to late filling of mitral inflow (E/A) were determined (**F**).

The online version of this article includes the following video, source data, and figure supplement(s) for figure 4:

**Figure supplement 1.** Heart functions are normal in *Ptpmt1$^{fl/fl}$/Ckm-Cre$^{+}$* mice at 7 months.

**Figure supplement 1—source data 1.** Uncropped immunoblotting images of *Figure 4—figure supplement 1F*.

**Figure 4—video 1.** Normal heart function in *Ptpmt1$^{+/+}$/Ckm-Cre$^{+}$* mice.

https://elifesciences.org/articles/86944/figures#fig4video1

**Figure 4—video 2.** Cardiac dysfunction in *Ptpmt1$^{fl/fl}$/Ckm-Cre$^{+}$* mice.

https://elifesciences.org/articles/86944/figures#fig4video2

---

arrhythmias, and impaired myocardial contraction (*Figure 5D–J*, *Figure 5—video 1*, and *Figure 5—video 2*). Cardiac strain images demonstrated that the global function of *Ptpmt1$^{fl/fl}$/Myh6-Cre$^{+}$* hearts, as reflected by global longitudinal strain, was significantly decreased (*Figure 5—figure supplement 2A, B*), similar to that observed in *Ptpmt1$^{fl/fl}$/Ckm-Cre$^{+}$* mice at the same age.

Unlike those in younger knockout mice, total cellular ATP levels in 10- to 12-month-old *Ptpmt1$^{fl/fl}$/Myh6-Cre$^{+}$* hearts were decreased by half (*Figure 6A*), and the knockout cardiomyocytes exhibited pronounced energetic stress as demonstrated by a substantial increase in p-Ampk (T$^{172}$) and decrease in p-mTor (S$^{2448}$) (*Figure 6B*). To determine the direct effects of Ptpmt1 loss on cardiomyocyte metabolism, we examined heart tissues isolated from young (3 months old) knockout mice before cardiomyocyte and mitochondria damages occurred. The mitochondrial content in these young *Ptpmt1* knockout cardiac tissues was comparable to that in control tissues (*Figure 6—figure supplement 1A*), and there were no changes in the expression levels of mitochondrial complexes (*Figure 6—figure supplement 1B*). Moreover, no bioenergetic/metabolic stress was detected in the knockout cardiomyocytes at this age (*Figure 6—figure supplement 1C*). However, like *Ptpmt1* knockout skeletal muscle mitochondria (*Figure 3B- D*), Ptpmt1-ablated cardiac mitochondria also showed altered fuel selection – the utilization of pyruvate was decreased (*Figure 6C*), whereas the free fatty acid and glutamate utilization was increased in *Ptpmt1$^{fl/fl}$/Myh6-Cre$^{+}$* cardiac mitochondria (*Figure 6D, E*). Intramitochondrial pyruvate levels in Ptpmt1-ablated cardiac mitochondria decreased (*Figure 6F*) while acetyl-CoA levels were not changed (*Figure 6—figure supplement 1D*), consistent with the metabolite data obtained from Ptpmt1-depleted skeletal muscle mitochondria (*Figure 3F, H*). Furthermore, acute production of α-KG from extramitochondrial pyruvate was decreased in the cardiac mitochondria lacking Ptpmt1 (*Figure 6G*), although Pdh activity in these mitochondria was not significantly affected (*Figure 6H*).

Oxidation of fatty acids in mitochondria yields much more ATP relative to carbohydrates (via pyruvate); however, fatty acid oxidation requires a greater rate of oxygen consumption for a given rate of ATP synthesis than carbohydrates. Such that the byproduct reactive oxygen species (ROS) are increased when mitochondria switch to fatty acids for energy metabolism. Indeed, *Ptpmt1* knockout heart tissues showed elevated overall cellular ROS levels even at 3 months (*Figure 6I*). Persistent oxidative stress can cause mitochondrial damage. Electron microscopic examination revealed that the myocardium of *Ptpmt1$^{fl/fl}$/Myh6-Cre$^{+}$* mice was distinguishable from that of *Ptpmt1$^{+/+}$/Myh6-Cre$^{+}$* control mice by the structures of mitochondria at 6–8 months (*Figure 6J*). Similar to that observed in *Ptpmt1* knockout skeletal muscle cells (*Figure 2C*), drastic accumulation of lipids or lipid intermediates was observed in 10- to 12-month-old *Ptpmt1$^{fl/fl}$/Myh6-Cre$^{+}$* cardiomyocytes (*Figure 6K*), which is likely responsible for the increased apoptosis and fibrillation in the knockout heart (*Figure 5B, C*) due to the toxicity of excess lipids (lipotoxicity) (*Goldberg et al., 2012*; *Huss and Kelly, 2005*).

## *Ptpmt1* is dispensable for the liver and adipose tissues

Finally, to further determine whether Ptpmt1-facilitated carbohydrate oxidation in mitochondria is also important for other metabolic tissues, we generated liver-specific (*Ptpmt1$^{fl/fl}$/Alb-Cre$^{+}$*) and fat-specific (*Ptpmt1$^{fl/fl}$/Adipoq-Cre$^{+}$*) *Ptpmt1* knockout mice by crossing *Ptpmt1 floxed* mice (*Yu et al., 2013*) with *Alb-Cre* (*Postic et al., 1999*) and *Adipoq-Cre* (*Eguchi et al., 2011*) mice, which express Cre specifically in hepatocytes and adipocytes, respectively. In contrast to muscle/heart- and heart-specific knockout

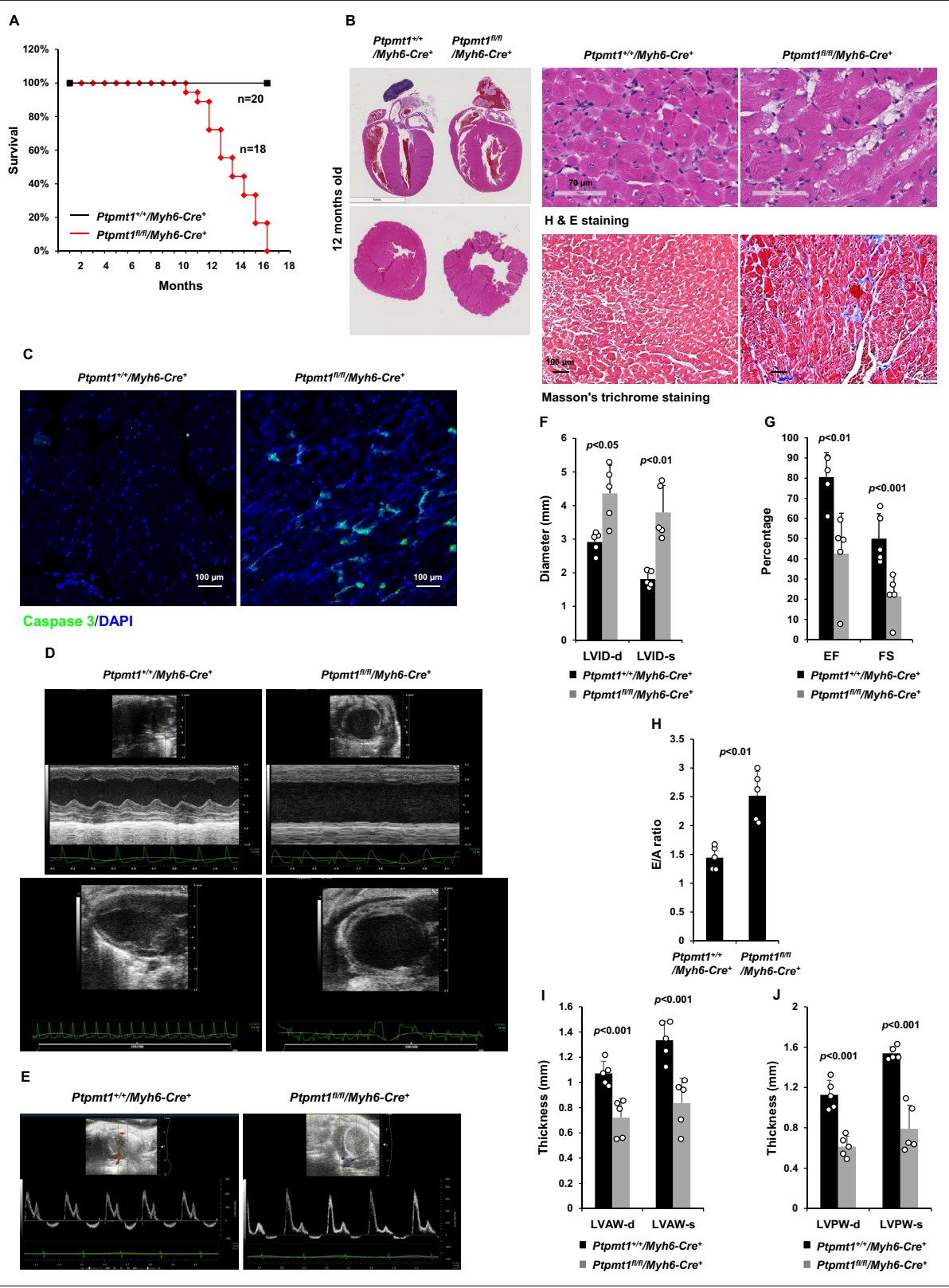

**Figure 5.** Deletion of *Ptpmt1* from the heart ultimately leads to dilated cardiomyopathy and heart failure. (**A**) Kaplan–Meier survival curves of *Ptpmt1^fl/fl^/ Myh6-Cre^+^* (*n* = 18) and *Ptpmt1^+/+^/Myh6-Cre^+^* mice (*n* = 20). (**B**) Heart tissue sections prepared from 10- to 12-month-old *Ptpmt1^fl/fl^/Myh6-Cre^+^* and *Ptpmt1^+/+^/Myh6-Cre^+^* mice (*n* = 4/genotype) were processed for Hematoxylin and Eosin (H&E) staining and Masson's Trichrome staining. Representative images are shown. (**C**) Heart tissue sections prepared from 11-month-old *Ptpmt1^fl/fl^/Myh6-Cre^+^* and *Ptpmt1^+/+^/Myh6-Cre^+^* mice were processed for

*Figure 5 continued on next page*

*Figure 5 continued*

immunofluorescence staining for cleaved caspase 3 followed by DAPI counterstaining. One representative image from 3 mice/genotype is shown. (**D–J**) Eleven- to twelve-month-old *Ptpmt1^fl/fl^/Myh6-Cre^+^* and *Ptpmt1^+/+^/Myh6 -Cre^+^* mice (*n* = 5/genotype) were examined by echocardiographic evaluation of ventricular function. Representative long- (upper panel) and short- (lower panel) axis views in M-mode echocardiographic tracing of left ventricle (LV) (**D**), and pulsed-wave Doppler recordings of mitral valve inflow (**E**) are shown. Left ventricular internal diameters at diastole (LVID-d) and systole (LVID-s) were measured (**F**). LV ejection fraction (EF) and fractional shortening (FS) (**G**), ratios of peak velocity of early to late filling of mitral inflow (E/A) (**H**), the thickness of LV anterior wall at the end of diastole (LVAW-d) and the end of systole (LVAW-s) (**I**), and thickness of LV posterior wall at the end of diastole (LVPW-d) and at the end of systole (LVPW-s) (**J**) were determined.

The online version of this article includes the following video and figure supplement(s) for figure 5:

**Figure supplement 1.** Cardiac functions are normal in middle-aged or younger heart-specific *Ptpmt1* knockout mice.

**Figure supplement 2.** Cardiac dysfunction in 12-month-old *Ptpmt1^fl/fl^/Myh6-Cre^+^* and *Ptpmt1^fl/fl^/Ckm-Cre^+^* mice.

**Figure 5—video 1.** Normal heart function in *Ptpmt1^+/+^/Myh6-Cre^+^* mice.

https://elifesciences.org/articles/86944/figures#fig5video1

**Figure 5—video 2.** Heart failure in *Ptpmt1^fl/fl^/Myh6-Cre^+^* mice.

https://elifesciences.org/articles/86944/figures#fig5video2

mice, liver-specific *Ptpmt1* knockout mice were healthy without overt abnormalities. They had a normal lifespan. Their body weights, liver weights, and blood glucose levels under both feeding and fasting conditions were comparable to those in control animals (*Figure 6—figure supplement 2A–D*). More-over, no significant differences were observed in glucose tolerance tests between *Ptpmt1^fl/fl^/Alb-Cre^+^* and control mice (*Figure 6–figure supplement 2E*). Unlike muscle/heart- and heart-specific *Ptpmt1* knockout mice in which prominent lipid accumulation was detected in muscle and cardiac myocytes, liver-specific *Ptpmt1* knockout mice did not display this secondary effect in hepatocytes, and there was no evident tissue damage in the liver or a fatty liver phenotype (*Figure 6—figure supplement 2F*). Similarly, adipocyte-specific *Ptpmt1* knockout mice were also healthy. Their body weights and glucose tolerance were comparable to those of control animals (*Figure 6—figure supplement 3A, B*). No histological changes were observed in *Ptpmt1* knockout white adipose and brown adipose tissues (*Figure 6—figure supplement 3C*). The fact that *Ptpmt1^fl/fl^/Alb-Cre^+^* and *Ptpmt1^fl/fl^/Adipoq-Cre^+^* mice well-tolerated Ptpmt1 deficiency in hepatocytes or adipocytes suggests that the detrimental effects of Ptpmt1 depletion-induced mitochondrial substrate shift are highly cell type specific.

## Discussion

Mitochondrial flexibility in substrate selection is essential for maintaining cellular energy homeostasis in physiology and under stress; however, the regulatory mechanisms and its significance for various tissues and cell types remain poorly characterized (*Smith et al., 2018*). In this study, we have identified an important role of the mitochondrial phosphatase Ptpmt1 in maintaining mitochondrial flexibility. The loss of Ptpmt1 decreased mitochondrial utilization of pyruvate, a key mitochondrial substrate derived from glucose, whereas fatty acid oxidation was enhanced (*Figure 3C* and *Figure 6D*). Impor-tantly, this prolonged mitochondrial substrate shift causes oxidative stress and mitochondrial damage, ultimately leading to the accumulation of lipids/lipid intermediates in *Ptpmt1* knockout skeletal muscle and cardiac myocytes. Excess lipids/lipid intermediates are known to be toxic to cardiomyocytes (lipo-toxicity) (*Goldberg et al., 2012*; *Huss and Kelly, 2005*). The profound damages in *Ptpmt1* knockout hearts and muscles that occurred subsequently were likely attributed to these accumulated lipids, although other factors, such as decreased anabolic activities (as demonstrated by diminished mTor signaling) (*Figure 2F* and *Figure 6B*), could also contribute to these phenotypes. The mitochondrial and tissue damages that subsequently happened, combined with the resulting energetic stress, finally led to muscle atrophy and heart failure in the knockout mice.

One of the important findings in this report is that persistent mitochondrial fuel switch from pyru-vate to fatty acids is detrimental for both cardiac and skeletal muscles despite the two tissues having different preferred energy sources. Glucose is the major energy source for skeletal muscles. It is used for energy production mainly in mitochondria in oxidative red muscles or through cytosolic glycolysis in glycolytic white muscles. Glucose is also the main energy source for the developing heart, but the adult heart primarily utilizes fatty acids to produce ATP, although the adult heart can also utilize glucose and lactate, the end product of glycolysis, as energy sources. Indeed, when mitochondrial utilization

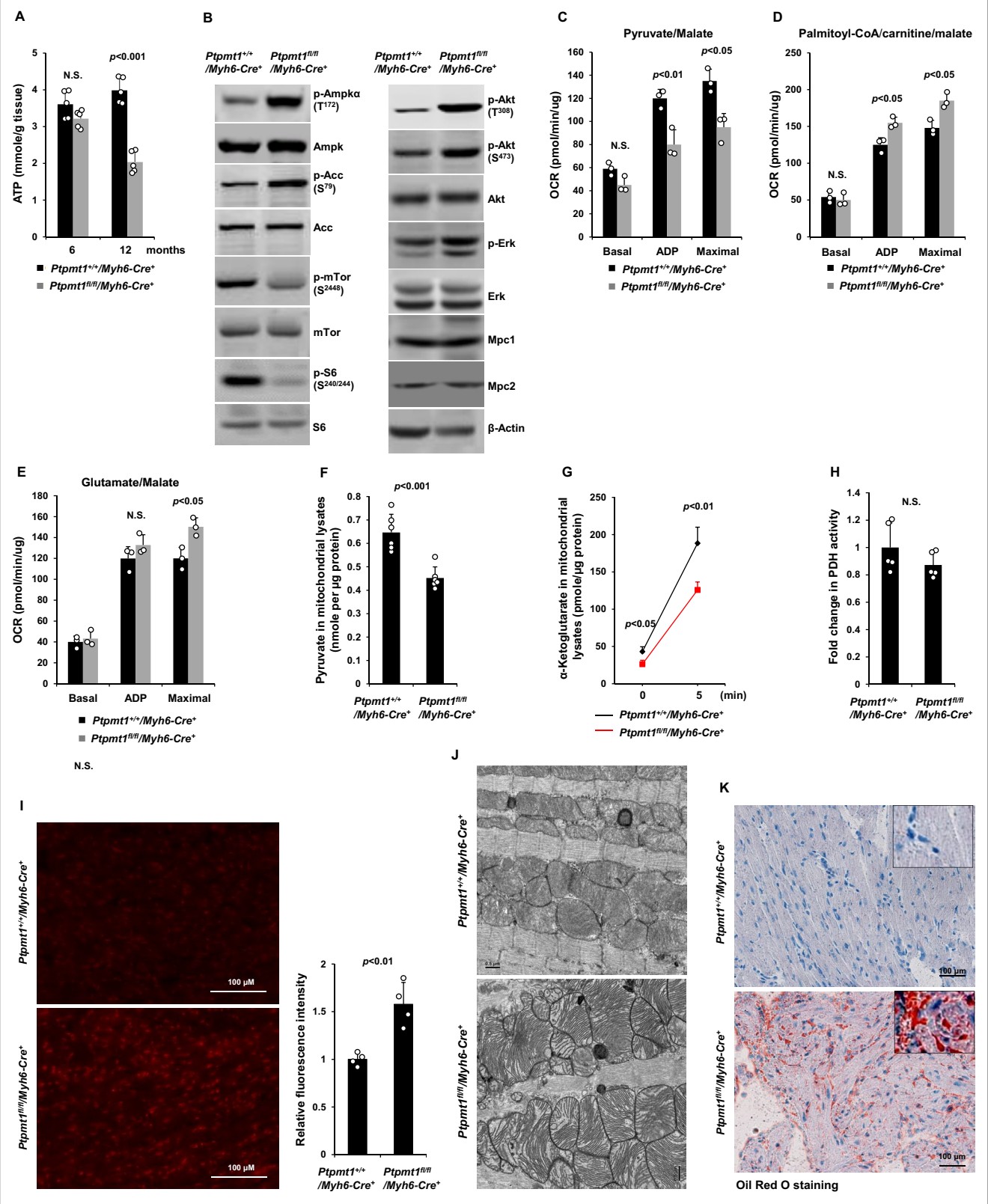

**Figure 6.** Ptpmt1 deficiency causes mitochondrial substrate shift and oxidative stress, leading to mitochondrial damage and lipid accumulation in *Ptpmt1* knockout cardiomyocytes. (**A**) Total ATP levels in the heart tissues dissected from *Ptpmt1*<sup>fl/fl</sup>/*Myh6-Cre*<sup>+</sup> and *Ptpmt1*<sup>+/+</sup>/*Myh6-Cre*<sup>+</sup> mice at the indicated ages (n = 5 mice/genotype). (**B**) Whole cell lysates prepared from the heart tissues of 10- to 12-month-old *Ptpmt1*<sup>fl/fl</sup>/*Myh6-Cre*<sup>+</sup> and *Ptpmt1*<sup>+/+</sup>/*Myh6-Cre*<sup>+</sup> mice were examined by immunoblotting with the indicated antibodies. Representative results from 3 mice/genotype are shown.

*Figure 6 continued on next page*

*Figure 6 continued*

Oxygen consumption of the mitochondria isolated from the heart tissues of 2- to 3-month-old *Ptpmt1^{fl/fl}/Myh6-Cre^+* and *Ptpmt1^{+/+}/Myh6-Cre^+* mice (*n* = 3/genotype) was measured in the presence of pyruvate (5 mM)/malate (5 mM) (**C**), palmitoyl-CoA (40 µM)/carnitine (40 µM)/malate (5 mM) (**D**), or glutamate (5 mM)/malate (5 mM) (**E**), following the addition of ADP (4 mM), oligomycin (1.5 µM), FCCP (4 µM), and antimycin A/rotenone (1 µM). OCRs at basal levels, in response to ADP addition, and maximal reserve capabilities were determined. (**F**) Levels of pyruvate in the lysates of the cardiac mitochondria isolated from 2- to 3-month-old *Ptpmt1^{fl/fl}/Myh6-Cre^+* and *Ptpmt1^{+/+}/Myh6-Cre^+* mice (*n* = 6/genotype) were determined. (**G, H**) Mitochondria isolated above were washed three times in Mitochondrial Assay Solution (MAS) buffer and then incubated with pyruvate (5 mM)/ malate (5 mM) and ADP (4 mM) at 37°C. Five minutes later, the mitochondria were collected, washed, and lysed. α-Ketoglutarate (α-KG) levels in the mitochondrial lysates were measured (*n* = 4 mice/genotype) (**G**). Pyruvate dehydrogenase (Pdh) activities in the mitochondrial lysates were determined (*n* = 5 mice/genotype) (**H**). (**I**) Heart tissue sections prepared from 3-month-old *Ptpmt1^{fl/fl}/Myh6-Cre^+* and *Ptpmt1^{+/+}/Myh6-Cre^+* mice (4 mice/genotype) were processed for reactive oxygen species (ROS) staining. One representative picture from 4 mice/genotype is shown. (**J**) Heart tissues dissected from 6- to 8-month-old *Ptpmt1^{fl/fl}/Myh6-Cre^+* and *Ptpmt1^{+/+}/ Myh6-Cre^+* mice were processed for transmission electron microscopic (TEM) examination. One representative image from 4 mice/genotype is shown. (**K**) Heart tissue sections prepared from 10- to 12-month-old *Ptpmt1^{fl/fl}/Myh6-Cre^+* and *Ptpmt1^{+/+}/ Myh6-Cre^+* mice were processed for Oil Red O staining to visualize lipids. One representative picture from 3 mice/genotype is shown.

The online version of this article includes the following source data and figure supplement(s) for figure 6:

**Source data 1.** Uncropped immunoblotting images of *Figure 6B*.

**Figure supplement 1.** There is no bioenergetic stress in young *Ptpmt1* knockout cardiomyocytes.

**Figure supplement 1—source data 1.** Uncropped immunoblotting images of *Figure 6—figure supplement 1C*.

**Figure supplement 2.** The impact of *Ptpmt1* deletion from the liver is minimal.

**Figure supplement 3.** No defects are observed in adipocyte-specific *Ptpmt1* knockout mice.

of pyruvate was inhibited by Ptpmt1 depletion, heart dysfunction developed much later than skeletal muscle dysfunction. Interestingly, the data from these animal models suggest that efficient carbohydrate oxidation in mitochondria and balanced mitochondrial fuel selection are critically important for both oxidative and glycolytic muscles as well as the adult heart, but not for the development of the skeletal muscle and heart although Ptpmt1 depletion took place during the embryonic stage. Undoubtedly, the late-onset phenotypes observed in the knockout mice over time was attributed to the metabolic defects arising from the loss of Ptpmt1 in the embryos. Although *Ptpmt1* knockout muscle cells and cardiomyocytes initially maintained energy homeostasis through enhanced fatty acid and glutamate oxidation, along with metabolic adaptations or activation of alternative energy-producing pathways in the first few months, they eventually encountered substantial energy deficits. This was attributed to the subsequent occurrence of oxidative stress and mitochondrial damage.

*Ptpmt1* knockout hepatocytes or adipocytes did not display such secondary effects (mitochondria and cell damages). This is possibly because the requirement of carbohydrate oxidation for energy production in hepatocytes and adipocytes is lower than that in skeletal muscle cells and cardiomyocytes, and/or *Ptpmt1* knockout hepatocytes and adipocytes did not need to uptake excessive fatty acids in the effort to maintain energy homeostasis. Another possibility is that the capabilities of hepatocytes and adipocytes to process fatty acids and to clear lipid metabolites (through cholesterol synthesis and cholesterol exportation out of the cell) are much stronger than muscle cells and cardiomyocytes (*Sinha et al., 2018*).

Another important finding in this report is that Ptpmt1 plays a major role in governing the metabolic fate of pyruvate and maintaining mitochondrial flexible substrate utilization. Pyruvate is at the central branch point of mitochondrial oxidation and cytosolic glycolysis in glucose metabolism – it can enter the mitochondrion for oxidation or be reduced to lactate in the cytosol. Our data suggest that inhibited mitochondrial utilization of pyruvate is most likely responsible for the remodeled metabolism in *Ptpmt1* knockout cells since ATP synthesis-driven oxygen consumption of Ptpmt1-ablated mitochondria was greatly reduced with pyruvate as the sole substrate (*Figure 3B* and *Figure 6C*). Moreover, acute production of α-KG from extramitochondrial pyruvate was indeed decreased (*Figure 3I*, *Figure 6G*).

The precise mechanism by which Ptpmt1 facilitates mitochondrial utilization/uptake of pyruvate remains to be further determined. Ptpmt1 is localized to the inner mitochondrial membrane where pyruvate transporters reside. This phosphatase dephosphorylates PIPs (*Yu et al., 2013*; *Shen et al., 2009*; *Pagliarini et al., 2004*). We have shown previously that PIPs directly promote fatty acid-induced activation of Ucp2 (*Yu et al., 2013*). Ucp2, unlike Ucp1, is a C4 metabolite transporter (*Vozza et al., 2014*). Elevated Ucp2 activity inhibits mitochondrial oxidation of glucose but enhances cytosolic

glycolysis although the underlying mechanisms remain unclear (*Bouillaud, 2009*; *Diano and Horvath, 2012*; *Pecqueur et al., 2008*; *Samudio et al., 2009*). Ptpmt1 depletion conceivably leads to the buildup of downstream PIP substrates in the mitochondrial inner membrane, which in turn remodel mitochondrial energy source selection by enhancing Ucp2 function (*Yu et al., 2013*) [Ucp2 levels in Ptpmt1-ablated mitochondria were not changed (*Figure 2—figure supplement 1A*, *Figure 6—figure supplement 1B*)]. It may also be possible that the accumulated PIPs directly inhibit the mitochondrial pyruvate transporter Mpc. It is important to note that Ptpmt1 was reported to be involved in the synthesis of cardiolipin (*Zhang et al., 2011*) which is one of the major components of the mitochondrial inner membrane and that plays an important role in maintaining mitochondrial membrane stability and dynamics. Thus, it may also be possible that Ptpmt1 depletion decreases mitochondrial pyruvate utilization through reduced cardiolipin synthesis. Comprehensive metabolomic analyses for *Ptpmt1*-deleted cells would provide additional insights into the mechanisms underlying the metabolic alterations caused by Ptpmt1 loss.

In summary, in this report we provide evidence that the mitochondria-based multifunctional phosphatase Ptpmt1 plays an important role in facilitating mitochondrial utilization of pyruvate and maintaining mitochondrial flexibility for balanced substrate selection. Prolonged mitochondrial energy sources switch from carbohydrates to lipids such as that caused by Ptpmt1 depletion is detrimental to the skeletal muscle and heart but not the liver and adipose, ultimately leading to muscle atrophy and heart failure.

# Materials and methods

## Key resources table

| Reagent type (species) or resource | Designation | Source or reference | Identifiers | Additional information |
|---|---|---|---|---|
| Antibody | anti-phospho-Ampkα (Thr$^{172}$) (40H9) (Rabbit monoclonal) | Cell Signaling Technology | Cat# 2535 | WB (1:1000) |
| Antibody | anti-Ampkα (D5A2) (Rabbit monoclonal) | Cell Signaling Technology | Cat# 5831 | WB (1:1000) |
| Antibody | anti-phospho-acetyl-CoA carboxylase (Ser$^{79}$) (D7D11) (Rabbit monoclonal) | Cell Signaling Technology | Cat# 11818 | WB (1:1000) |
| Antibody | anti-acetyl-CoA carboxylase (C83B10) (Rabbit monoclonal) | Cell Signaling Technology | Cat# 3676 | WB (1:1000) |
| Antibody | anti-phospho-mTor (Ser$^{2448}$) (D9C2) (Rabbit monoclonal) | Cell Signaling Technology | Cat# 5536 | WB (1:1000) |
| Antibody | anti-mTor (7C10) (Rabbit monoclonal) | Cell Signaling Technology | Cat# 2983 | WB (1:1000) |
| Antibody | anti-phospho-p70 S6 kinase (Thr$^{389}$) (108D2) (Rabbit monoclonal) | Cell Signaling Technology | Cat# 9234 | WB (1:1000) |
| Antibody | anti-p70 S6 kinase (49D7) (Rabbit monoclonal) | Cell Signaling Technology | Cat# 2708 | WB (1:1000) |
| Antibody | anti-phospho-S6 ribosomal protein (Ser$^{240/244}$) (D68F8) (Rabbit monoclonal) | Cell Signaling Technology | Cat# 5364 | WB (1:1000) |
| Antibody | anti-S6 ribosomal protein (5G10) (Rabbit monoclonal) | Cell Signaling Technology | Cat# 2217 | WB (1:1000) |
| Antibody | anti-β-actin (C-4) (Mouse monoclonal) | Santa Cruz Biotechnology | Cat# SC47778 | WB (1:1000) |
| Antibody | anti-phospho-4E-bp1 (Thr$^{37/46}$) (236B4) (Rabbit monoclonal) | Cell Signaling Technology | Cat# 2855 | WB (1:1000) |
| Antibody | anti-phospho-Akt (Thr$^{308}$) (D25E6) (Rabbit monoclonal) | Cell Signaling Technology | Cat# 13038 | WB (1:1000) |
| Antibody | anti-phospho-Akt (Ser$^{473}$) (D9E) (Rabbit monoclonal) | Cell Signaling Technology | Cat# 4060 | WB (1:1000) |

*Continued on next page*

*Continued*

| Reagent type (species) or resource | Designation | Source or reference | Identifiers | Additional information |
|---|---|---|---|---|
| Antibody | anti-Akt (pan) (C67E7) (Rabbit monoclonal) | Cell Signaling Technology | Cat# 4691 | WB (1:1000) |
| Antibody | anti-phospho-Erk (E-4) (Mouse monoclonal) | Santa Cruz Biotechnology | Cat# SC7383 | WB (1:1000) |
| Antibody | anti-p44/42 Mapk (Erk1/2) (Rabbit monoclonal) | Cell Signaling Technology | Cat# 9102 | WB (1:1000) |
| Antibody | anti-Mpc1 (D2L9I) (Rabbit monoclonal) | Cell Signaling Technology | Cat# 14462 | WB (1:1000) |
| Antibody | anti-Mpc2 (D4I7G) (Rabbit monoclonal) | Cell Signaling Technology | Cat# 46141 | WB (1:1000) |
| Antibody | anti-total OXPHOS rodent WB antibody cocktail (Mouse monoclonal): anti-NDUFB8 (20E9DH10C12), anti-SDHB (21A11AE7), anti-UQCRC2 (13G12AF12BB11), anti-MTCO1 (1D6E1A8), and anti-ATP5A (15H4C4) | Abcam | Cat# ab110413 | WB (1:1000) |
| Antibody | anti-Ucp2 Rabbit Ab (C-20) | Santa Cruz Biotechnology | Cat# SC6525 | WB (1:1000) |
| Antibody | anti-Lc3a (Rabbit monoclonal) | Novus Biologicals | Cat# NB100-2331 | WB (1:500) |
| Antibody | anti-cleaved caspase-3 (Asp175) (Rabbit monoclonal) | Cell Signaling Technology | Cat# 9661 | IF (1:400) |

## Mice

*Ptpmt1$^{fl/+}$* mice with the C57Bl/6 background were generated in our previous study (*Yu et al., 2013*). *Ckm-Cre$^+$* (*Brüning et al., 1998*) (Stock# 006475), *Myh6-Cre$^+$* (*Agah et al., 1997*) (Stock# 011038), *Alb-Cre$^+$* (*Postic et al., 1999*) (Stock# 003574), and *Adipoq-Cre$^+$* (*Eguchi et al., 2011*) (Stock# 010803) mice with the C57BL/6 background were purchased from the Jackson Laboratory. To avoid potential side effects of Cre, *Ptpmt1$^{+/+}$/Cre$^+$* were used as controls for *Ptpmt1$^{fl/fl}$/Cre$^+$* and the *Cre* transgene was hemizygous in both mouse types. In initial pilot experiments, we examined *Ptpmt1$^{+/+}$/Cre$^+$* and no defects were observed in these animals. Mice were kept under specific pathogen-free conditions in Case Western Reserve University Animal Resources Center and Emory University Division of Animal Resources (Protocol # PROTO201700884). Mice of the same age, sex, and genotype were randomly grouped for subsequent analyses (investigators were not blinded). All animal procedures complied with the NIH Guidelines for the Care and Use of Laboratory Animals and were approved by the Institutional Animal Care and Use Committee.

## Antibodies

The sources and catalog numbers of the antibodies utilized in this study are listed in the Key Resources Table.

## Wire hang test

Maximal muscular strength of mouse forelimbs was assessed by wire hang tests following a standard protocol. Mice were allowed to grasp and hang on a wire with forelimbs. Time duration when the mouse was able to suspend before falling was recorded. Three trials were carried out for each mouse. The average time was normalized against body weight. Values of control mice were set to 100%.

## Treadmill exercise test

Treadmill exercise tests were conducted following the protocol previously reported by trained personnel at Case Western Reserve University Mouse Metabolic Phenotyping Center Analytical Core (*Desai et al., 1997*; *Shechtman and Talan, 1994*). Adjustable parameters on the treadmill include belt speed (0–99 m/min) and angle of inclination (0–45"). After acclimation for 1 hr, 2.5 m/min incremental increases in treadmill belt speed and 2' increments in an angle of inclination were performed every

3 min until the mouse exhibited signs of exhaustion. Exhaustion was defined as the mouse spending >50% of the time or X5 sec consecutively on the shock grid. The maximum speed and duration of the run were recorded at the time of exhaustion.

## Ex vivo muscle contractility assay

Intact Soleus and EDL isolated from mice were used for the collection of isometric force data using an eight-chamber system (*Shen et al., 2009*; *Thornton et al., 2011*). Data were analyzed with ADInstruments PowerLab software customized for these experiments. Muscles were first equilibrated for 20 min to mimic conditions of normal activity (low duty cycle, ~1%). Muscles were then subjected to the length–force relationship test to determine the optimal length at which maximal force is achieved. Muscles were stimulated with frequencies ranging from 1 to 130 Hz to generate force versus frequency relationships. From these relationships, the frequency producing maximal tetanic force ($T_{max}$) was then used for the rest of the experiments. Muscles were stimulated every minute with $T_{max}$ for 5–10 min to assure that the preparation was stable at which point the tissue bathing buffer was replaced with one containing 0 mM $Ca^{2+}$ + 0.1 mM Ethylene glycol tetraacetic acid (EGTA). The muscles were then stimulated every minute at $T_{max}$ to determine the impact of the removal of external calcium on muscle force. After 20 min, the 0 mM $Ca^{2+}$ buffer was washed out and replaced by a buffer containing 2.5 mM $Ca^{2+}$, and the muscles were stimulated every minute for 20 min at $T_{max}$ to allow for force recovery. After force measurement protocols, muscle dimensions and masses were measured for the determination of cross-sectional normalized forces (relative force). Muscle force was reported as absolute force (mN) and force normalized to the following muscle physiological cross-sectional area (PCSA, N/$cm^2$) equation as previously reported (*de Paula Brotto et al., 2001*; *Park et al., 2012*) with a small modification in that in our final length calculation, we considered the length to weight ratio of muscle fibers for stretched muscles (tendon-to-tendon) of ex vivo muscles in contractility chambers (*Kitase et al., 2018*) as the actual muscle fiber size was smaller than the length of the measured stretched muscle. Since the PCSA is very sensitive to length, by adjusting to the length of the muscle fiber, force estimation was more precise. Thus, our final formula was: Muscle force (N/$cm^2$) = (force (g) × muscle length (cm) × 0.75 (EDL) or 0.85 (Soleus) × 1.06 (muscle density))/(muscle weight (g) × 0.00981).

## Echocardiography

Echocardiographic studies were performed by trained investigators at Emory University Department of Pediatrics Animal Physiology Core as previously reported (*Winterberg et al., 2016*; *Agarwal et al., 2016*). Mice were anesthetized with 1–2% isoflurane/100% oxygen continuously and positioned on a warming platform set to 37°C. The heart rate was controlled at 360–460 beats per minute to standardize recordings. ECG and respiratory rate were monitored for physiological assessment while scanning. A Vevo 2100 digital high-frequency ultrasound system (FujiFilm Visualsonics Inc, Toronto, ON, Canada) equipped with an MS400, 30-MHz transducer was used to acquire the images. Interventricular septal thickness (IVS), left ventricular internal dimensions (LVID), and posterior wall thickness (LVPW) at diastole and systole (IVS-d, LVID-d, LVPW-d and IVS-s, LVID-s, LVPW-s) were measured from M-mode images in both parasternal long- and short-axis views. Left ventricle volume at diastole and systole (LV Vol-d, LV Vol-s), left ventricular ejection fraction (LVEF), and left ventricular fractional shortening (LVFS) were calculated as the following formula (Vevo 2100 Workstation Software VisualSonics Inc, Toronto, ON, Canada): LV Vol-d(ul) = [7.0/(2.4 + LVID-d)] * LVID-d3; LV Vol-d(ul) = [7.0/(2.4 + LVID-s)] * LVID-s3; EF (%) = 100 * [(LV Vol-d − LV Vol-s)/LV Vol-d]; FS (%) = 100 * [(LVID-d − LVID-s)/LVID-d]. The mitral valve flow Doppler velocities were measured by the apical four-chamber view. Early (E) and late (A) mitral valve inflow velocity and mitral valve E to A ratios (MV E/A) were also obtained.

Echocardiographic speckle-tracking-based strain measurement was also performed using the VevoStrain application (VisualSonics Vevo 2100 Imaging System, FujiFilm VisualSonics, Inc, Toronto, ON, Canada) which produces velocity strain and time-to-peak analyses on myocardial wall images. The B-mode cine loop was acquired for Vevo Strain analysis. The qualified consecutive cardiac cycles were selected between one respiration cycle and the next in the Electrocardiogram (EKG) panel. Both endocardium and epicardium were traced simultaneously and semi-automatically at the parasternal long-axis view. VevoStrain builds the dynamic LV wall trace for all frames in the cine loop. Velocity, displacement, strain, and strain rate data were calculated based on a speckling-tracking algorithm

provided by the Vevo software. Data on overall cardiac tissue performance, segmental time-to-peak, and segmental phase quantification were also collected.

## Intracellular ATP quantification

Skeletal muscle and heart tissues were lysed in lysis buffer [Tris–HCl (50 mM), pH 7.4, NP-40 (1%), Na-deoxycholate (0.25%), NaCl (150 mM), Ethylenediamine tetraacetic acid (EDTA) (1 mM), NaF (1 mM), $Na_3VO_4$ (1 mM), Phenylmethylsulfonyl fluoride (PMSF) (1 mM), and aprotinin/leupeptin (10 µg/ml)] for 30 min. Samples were then centrifuged (10,000 × $g$) for 10 min, and the supernatants were collected. Total ATP levels were assessed using an ATP bioluminescence assay kit (Sigma, St. Louis, MO), following the instructions provided by the manufacturer. ATP levels were normalized against tissue weights or protein contents in the samples.

## Measurement of respirationin isolated mitochondria

Mitochondria were freshly isolated from skeletal muscle and heart tissues. Respiration of mitochondria was measured using a Seahorse XF24 analyzer as described previously (*Rogers et al., 2011*). Mitochondria (10 µg of protein) were plated in each well of a XF24 plate in 50 µl 1× Mitochondrial Assay Solution (MAS), pH 7.4 [sucrose (70 mM), mannitol (220 mM), $KH_2PO_4$ (5 mM), $MgCl_2$ (5 mM), 4-(2-hydroxyethyl)-1-piperazineethanesulfonic acid (HEPES) (2 mM), EGTA (1 mM), and fatty acids-free bovine serum albumin (0.2%)]. XF24 plates were centrifuged at 4°C for 20 min at 2000 × $g$, and then 450 µl of MAS containing 5 mM pyruvate/5 mM malate, 5 mM glutamate/5 mM malate, 40 µM palmitoyl-CoA/40 µM carnitine/5 mM malate, or 10 mM succinate was added to each well. Plates were incubated at 37°C in an incubator for 8–10 min. Oxygen consumption rates (OCRs) were measured under basal conditions in the presence of ADP (4 mM), the ATP synthase inhibitor oligomycin (1.5 µM), the mitochondrial uncoupling compound carbonylcyanide-4-trifluorometh-oxyphenylhydrazone (FCCP, 4 µM), and the electron transport chain inhibitor rotenone (1 µM).

## Measurement of respiration in fresh muscle tissues

OCRs of muscle tissues were measured using a Seahorse XF24 analyzer and Seahorse XF24 Islet Capture Microplates (Agilent, 101122-100) as described previously (*Shintaku et al., 2016*). Briefly, tibialis anterior muscles were dissected into 3 × 1 mm cross-sections with a biopsy punch (3 mm diameter). The tissue sections were incubated for 1 hr in Dulbecco's modified Eagle medium (DMEM) supplemented with 10 mM glucose, 2 mM l-glutamine, and 1 mM sodium pyruvate. Tissue OCRs were measured at the basal level and following the addition of oligomycin (8 µM), FCCP (4 µM), and antimycin A/rotenone (1 µM). Following OCR measurements, tissues were dried at 60°C for 48 hr. All tissue OCR measurements were then normalized against dry tissue weights.

## Mitochondrial Pdh activity assay, pyruvate, acetyl-CoA, and α-KG measurements

Mitochondria were freshly isolated from skeletal muscle/heart tissues and washed twice with MAS. Mitochondrial pellets were lysed in [Tris–HCl (pH 7.4, 50 mM), NP-40 (1%), Na-deoxycholate (0.25%), NaCl (150 mM), EDTA (1 mM), NaF (1 mM), $Na_3VO_4$ (1 mM), PMSF (1 mM), aprotinin/leupeptin (10 µg/ml)]. Lysates were subjected to the Pdh activity assay and pyruvate, acetyl-CoA, and α-KG measurements using Pdh activity colorimetric assay, pyruvate assay, acetyl-CoA assay, and α-KG assay kits purchased from BioVision following the manufacturer's instructions.

## Histological and transmission electron microscopic examination

Freshly dissected skeletal muscle and heart tissues were embedded in Optimal Cutting Temperature (OCT) compound (Tissue-Tek) and frozen at −80°C. Slides were stained for Hematoxylin and Eosin (H&E), Masson's Trichrome, and Oil Red O. For transmission electron microscopic (TEM) analyses, mice were perfused with 3% paraformaldehyde, 1.5% glutaraldehyde, 100 mM cacodylate (pH 7.4), and 2.5% sucrose. Skeletal muscle and heart tissues were postfixed for 1 hr and then processed for TEM examination.

## Tissue ROS determination

Freshly dissected heart tissues were embedded in OCT compound (Tissue-Tek) and immediately placed on crushed dry ice. Frozen sections were prepared at 8 µm thickness. After being rinsed in

pure $H_2O$ for 30 s to wash out OCT compound, slides were stained with 5 µM dihydroethidium (DHE) at room temperature in dark for 15 min. Slides were washed with deionized $H_2O$ for 1 min three times and visualized immediately using a fluorescence microscope. DHE-derived $2\text{-OH-E}^+$, a specific adduct of cellular superoxide, was visualized with a red excitation filter. Equal exposure times were applied to all slides. Fluorescence intensities were measured by ImageJ software.

## Immunoblotting

Skeletal muscle and heart tissues were lysed in Radioimmunoprecipitation assay (RIPA) buffer (50 mM Tris–HCl pH 7.4, 1% NP-40, 0.25% Na-deoxycholate, 150 mM NaCl, 1 mM EDTA, 1 mM NaF, 1 mM $Na_3VO_4$, 10 µg/ml leupeptin, 10 µg/ml aprotinin, and 1 mM PMSF). Lysates (50 µg) were resolved by sodium dodecyl sulfate–polyacrylamide gel electrophoresis followed by immunoblotting with the indicated antibodies following standard procedures.

## Statistics

Data are presented as mean ± standard deviation of all mice analyzed (i.e., biological replicates) in multiple experiments. Tissues, mitochondria, etc. that were analyzed were isolated from single individual mice (not pooled). Statistical significance was determined using unpaired two-tailed Student's *t*-test. *p < 0.05; **p < 0.01; ***p < 0.001. N.S. indicates not significant.

## Acknowledgements

We are grateful for the technical assistance provided by Julian Vallejo and the services provided by Children's Healthcare of Atlanta and Emory University's Animal Physiology Core, and Case Mouse Metabolic Phenotyping Center. This work was supported by the National Institutes of Health grants HL130995 and HL162725 (to C.K.Q.).

## Additional information

### Funding

| Funder | Grant reference number | Author |
|---|---|---|
| National Institutes of Health | HL130995 | Hong Zheng<br>Qianjin Li<br>Shanhu Li<br>Zhiguo Li<br>Ashruth Reddy<br>Cheng-Kui Qu |
| National Institutes of Health | HL162725 | Hong Zheng<br>Qianjin Li<br>Shanhu Li<br>Zhiguo Li<br>Ashruth Reddy<br>Cheng-Kui Qu |

The funders had no role in study design, data collection, and interpretation, or the decision to submit the work for publication.

### Author contributions

Hong Zheng, Qianjin Li, Shanhu Li, Michelle Puchowicz, Data curation, Formal analysis, Investigation, Methodology, Writing - original draft, Writing – review and editing; Zhiguo Li, Marco Brotto, Domenick Prosdocimo, Data curation, Formal analysis, Investigation, Methodology, Writing – review and editing; Daiana Weiss, Ashruth Reddy, Investigation, Writing – review and editing; Chunhui Xu, Xinyang Zhao, Writing – review and editing; M Neale Weitzmann, Mukesh K Jain, Supervision, Writing – review and editing; Cheng-Kui Qu, Conceptualization, Supervision, Funding acquisition, Writing - original draft, Project administration, Writing – review and editing

### Author ORCIDs

Hong Zheng ⬆ http://orcid.org/0000-0002-4048-6530

Qianjin Li http://orcid.org/0009-0009-4993-0712
M Neale Weitzmann https://orcid.org/0000-0003-3305-5748
Cheng-Kui Qu http://orcid.org/0000-0002-4256-8652

### Ethics

All animal procedures complied with the NIH Guidelines for the Care and Use of Laboratory Animals and were approved by the Institutional Animal Care and Use Committee (Protocol # PROTO201700884).

Reviewer #1 (Public Review): https://doi.org/10.7554/eLife.86944.3.sa1
Reviewer #2 (Public Review): https://doi.org/10.7554/eLife.86944.3.sa2
Author Response https://doi.org/10.7554/eLife.86944.3.sa3

---

## Additional files

### Supplementary files

• MDAR checklist

### Data availability

All data generated or analyzed during this study are included in the manuscript and supporting files; Source Data files have been provided for Figure 1-figure supplement 2, Figure 2, Figure 2-figure supplement 1, Figure 4-figure supplement 1, Figure 6 and Figure 6-figure supplement 1.

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
