## [Editor Report · eLife assessment]

This paper provides a **useful** set of data examining the role of PTPMT1, a mitochondria-based phosphatase, in mitochondrial fuel selection. The data were collected and analyzed using **solid** methodology and can be used as a starting point for further studies that build on the findings here.

---

## [Referee Report · Reviewer #1 (Public Review)]

The manuscript by Zheng, et al., is focused on assessing the role of deletion of PTPMT1, a mitochondria-based phosphatase, in mitochondrial fuel selection. Authors show that the utilization of pyruvate, a key mitochondrial substrate derived from glucose, is inhibited, whereas fatty acid utilization is enhanced. Importantly, while the deletion of PTPMT1 does not impact development of skeletal muscle or heart, the metabolic inflexibility leads to muscular atrophy, heart failure, and sudden death. Mechanistically, authors claim that the prolonged substrate shift from carbohydrates to lipids causes oxidative stress and mitochondrial dysfunction, leading to accumulation of lipids and muscle cell and CM damage in the KO. Interestingly, PTPMT1 deletion from the liver or adipose tissue does not generate any local or systemic defects. Authors conclude that PTPMT1 plays an important role in maintaining mitochondrial flexibility and that the balanced utilization of carbohydrates and lipids is essential for skeletal muscle and heart.

The following issues remain:

1. Authors have not alleviated the concern regarding the fact that CKMM- and the MYHC-Cre express early, during development ; even if the effects are not grossly apparent during development, many developmental issues progress over time and manifest later in adulthood, particularly those concerning cardiac function and development (ie adult congenital disease). As such, the authors explanation that they don't observe differences does not suffice; detailed developmental assessment by histology at the various developmental stages (by timed mating) are needed to validate the study and conclusions of the authors. Alternatively, as mentioned previously, authors could utilize inducible cre drivers, expressing the gene only in adulthood to prove that the effects are or not developmental in nature. Similarly, the authors new assertion that late-onset phenotypes observed in the knockout mice over time is attributed to the metabolic defects arising from the loss of PTPMT1 in the embryos needs to be validated- therefore the developmental effects are in fact critical to the phenotype and should be demonstrated in the paper.

2. Quantification of ALL western blot data is an absolute necessity and speaks to the rigor and reproducibility of the study. I do not agree that this is unnecessary or that it would take up too much space.

---

## [Referee Report · Reviewer #2 (Public Review)]

This study presents novel findings on the metabolic fuel preference shift regulated by PTPMT1, a target of interest, in skeletal and cardiac muscle cells.

Zheng et al. have investigated the effects of PTPMT1 Knock-out on cellular metabolic flexibility. Since the authors used several types of appropriate tissue-specific mouse models, it seems to be a broad significance at the first glance. However, most of the data lack the quantification, consequently they don't provide statistical significance. In addition, the functional data such as echocardiography shows partial and limited data.

Therefore, it is only a matter of speculation that the absence of PTPMT1 inhibits glucose (pyruvate) utilization and promotes FAO.

---

## [Author Response]

The following is the authors’ response to the current reviews.

Comment 1: The descriptions about body weights should be matched.

Regrettably, we did not monitor the body weights throughout the study. We have now revised the description clarifying the confusions. Importantly we evaluated the weights of the muscle (EDL and soleus) and heart tissues in 8-month-old mice (Fig. 1A).

Comment 2: Quantitative data for figures.

As stated in the manuscript, the presented images are representatives of at least three mice per genotype. However, assessing specific measurements such as cell sizes, diameters, or mitochondria sizes in histological tissue sections and electron microscopical fields is not feasible due to practical limitations. Unfortunately, we do not have access to specialized software for such analyses. While semi-quantification of Western blot bands is possible, implementing this for all Western blots in the manuscript would result in a substantial increase in the number of bar graphics. Below are Western blots from additional two pairs of mice used in all figures.

Comment 3: Confusions about “total mitochondrial content”.

The mitochondria content in cells was assessed by quantitatively comparing the DNA level of the mitochondrial gene cytochrome B to that of the nuclear gene 18S using quantitative PCR. This method is commonly used to determine the relative number of mitochondria in cells. However, we have revised and provided a clearer description in the figure legend to avoid any potential confusion.

Comment 4: Suggestions on further analyses of PGC1-alpha and TFAM. LC3-I and -II.

We evaluated LC3-I/II levels in PTPMT1 knockout muscles, and our findings did not indicate any signs of increased autophagic activity (Supplementary Figure S3). We will examine PGC1-alph and TFAM levels in our future studies. It is worth noting that in our previous RNA-seq analyses of PTPMT1 knockout hematopoietic cells, we did not observe any significant alterations in the expression levels of these two genes.

Comment 5: Description on fibrotic lesions.

Quantifying fibrotic areas poses a significant challenge. Therefore, we were only able to describe this finding.

Comment 6: Fig 6 is not well organized and aligned.

In response to your suggestion, we have reorganized this figure accordingly. Panels C, D, and E display mitochondrial OCR data derived from three biological replicates/genotype. We feel that these changes are sufficient to demonstrate the differences in substrate utilization between PTPMT1 knockout and control mitochondria.

Comment 7: Descriptions on glucose oxidation and glycolysis in different types of muscle fibers are confusing

We have followed the suggestions and revised the descriptions accordingly.

Comment 8: A discussion about lactate utilization in cardiomyocytes would be helpful.

Following this suggestion, we have now added a brief discussion.

Comment 9: “Cropped” images were used in Fig 10.

The images shown in Fig. 10 were not cropped images. In order to efficiently use the tissue and mitochondrial lysates, the Western blot membranes were intentionally cut into smaller fragments based on the molecular weights of the proteins to be detected. These smaller membrane sections were then employed for individual Western blotting purposes.

Minor comment 1: The order of Fig 1 panels should be reorganized.

Following this suggestion, we have now reorganized this figure.

Minor comment 2: Suggestion for an Echocardiograph result table.

These analyses were carried out by trained personnel at the Emory Animal Physiology Core. The data presented in our manuscript was provided by them. It is important to note that no additional parameters were measured beyond the data provided by the Core.

Minor comment 3: Is ROS production increased in PTPMT1 knockout muscle cells?

Yes, PTPMT1 knockout tissues showed elevated overall cellular ROS levels even at 3 months (Figure 6I).

Minor comment 4: Typo in S10 legend.

The typo has been corrected.

The following is the authors’ response to the original reviews.

Comment 1: The effects of PTPMT1 on the skeletal muscle and heart might be an embryonic defect. They might be mediated by significantly reduced mTOR signaling

We acknowledge the valid point made by this reviewer. While both CKMM-Cre and Myh6Cre express Cre during the embryonic stage, we did not observe any developmental defects in skeletal muscle-specific (PTPMT1fl/fl/CKMM-Cre) or heart-specific (PTPMT1fl/fl/Myh6-Cre) knockout mice. These knockout mice appeared indistinguishable from their WT littermates until the age of 3-4 months.

Morphologically, the skeletal muscle and heart dissected from these mice showed no abnormalities.Additionally, mitochondria isolated from these tissues did not exhibit any morphological/structural defects. Undoubtedly, the late-onset phenotypes observed in the knockout mice over time was attributed to the metabolic defects arising from the loss of PTPMT1 in the embryos. Although PTPMT1 knockout muscle cells and cardiomyocytes initially maintained energy homeostasis through enhanced fatty acid and glutamate oxidation, along with metabolic adaptations or activation of alternative energy-producing pathways in the first few months, they eventually encountered substantial energy deficits. This was attributed to the subsequent occurrence of oxidative stress and mitochondrial damage. In response to this valuable feedback, we have included a brief discussion in the manuscript's discussion section to address this point.

As mentioned in the manuscript, the late-onset phenotypes observed in our study were likely a result of subsequent damages induced by prolonged metabolic substrate shift and lipid accumulation within the cells. We agree with the reviewer that decreased mTOR activities may also contribute to these late effects, and have included a brief discussion in the discussion section.

Comment 2: Why are the effects of the loss of PTPMT1 similar in the skeletal muscle and heart.

The depletion of PTPMT1 yields similar effects in both tissue types; however, the manifestations occur earlier in the skeletal muscle. Although mitochondria in the skeletal muscle and heart have distinct preferences for energy sources, prolonged forced utilization of fatty acids caused by PTPMT1 depletion eventually leads to lipid accumulation and cellular damage (lipotoxicity) in both tissue types. This phenomenon underscores the importance of maintaining a balance in substrate utilization to prevent adverse effects on cellular health in the skeletal muscle and heart.

Comment 3: AMPK is activated in PTPMT1 knockout cardiomyocytes; this should have cardioprotective effects.

AMPK can be activated through various mechanisms. In our study, AMPK activation occurs in response to energetic stress in late-stage PTPMT1 knockout tissues that displayed significantly reduced ATP levels, aligning with its role as a bioenergetic stress sensor. It is possible that AMPK activation alone was insufficient to overcome the secondary damages induced by the prolonged metabolic switch from carbohydrate metabolism to fatty acid metabolism.

Comment 4: Knockout skeletal muscles and hearts had lipid accumulation; why were knockout mice smaller than controls? Are there any changes in white fat, core temperature or browning of fat? Rescue experiments should be considered to prove that lipid accumulation is the cause of death in the knockout mice.

We believe that the lipid accumulation observed in muscle cells and cardiomyocytes of the knockout mice does not necessarily imply that these tissue-specific knockout mice would be heavier or have increased body fat. We appreciate the suggestions regarding energy expenditure tests and rescue experiments. We will certainly consider incorporating these experiments into our future study.

As stated in the manuscript, we did not observe any morphological changes in white or brown fat tissues in the adipocyte-specific PTPMT1 knockout mice. Furthermore, we assessed body temperature and its response to a cold environment (4°C), and no differences were detected between the knockout mice and the control mice.

Comment 5: Are there sex differences in muscle and heart phenotypes in the tissue specific knockout mice?

We did not observe significant differences in phenotypes between male and female knockout mice.

Comment 6: What happens to UCP2 activity in PTPMT1 deleted cells and what is its function in mediating AMPK and/mTOR regulation.

Currently, there is a lack of direct methods available to measure UCP2 activity. The relationship between UCP2 and the regulation of AMPK and mTOR has not been extensively investigated.

Comment 7: What is the effect of PTPMT1 deletion on cardiolipin synthesis?

PTPMT1 has been implicated in both facilitating mitochondrial utilization of pyruvate and participating in the synthesis of cardiolipin. To investigate the impact of PTPMT1 knockout on cardiolipin levels, we plan to establish a mass spectrometry assay for the quantitative analysis of cardiolipin in knockout mitochondria. Completing these experiments might require a considerable amount of time. Nonetheless, we extensively addressed this point in the discussion section.

Minor concerns:Comment 8: The title needs more specificity.

As suggested, we have revised the title to "Loss of PTPMT1 restricts mitochondrial utilization of carbohydrates and induces muscle atrophy and heart failure in tissue-specific knockout mice".

Comment 9: Heart and skeletal muscle weights in Fig 1A should be normalized against tibia length.

Unfortunately, we did not perform normalization in this study. However, we appreciate the suggestion and will incorporate it into our future studies. It is important to note that the lengths of tibias in the knockout mice were only marginally shorter.

Comment 10: Low magnification and longitudinal section of the muscle should be shown in Fig 1B and 2A.

The histological images provide supporting evidence for the conclusion, despite not being optimal in quality. We acknowledge the suggested improvements and assure you that we will integrate them into our future studies. It is crucial to emphasize that each conclusion in this study was derived from multiple experimental designs, rather than solely relying on morphological changes.

Comment 11: Fig 1F is mislabeled as 1G.

We have conducted a thorough review and can confidently confirm that the labeling is correct.

Comment 12: Fig 2F and 6B should be quantified.

As indicated in the manuscript, the images presented are representatives of at least three mice per genotype. While semi-quantification of Western blot bands is possible, implementing this for all Western blots in the manuscript would result in a substantial increase in the number of bar graphics.Below are Western blot images from additional two pairs of mice included in Fig. 2F and Fig. 6B. Furthermore, Western blot images from two additional pairs of mice in other figures are also provided below.

**Author response image 1. sa3fig1:** Western blotting data from additional two pairs of mice in Fig.2F.

**Author response image 2. sa3fig2:** Western blotting data from additional two pairs of mice in Fig.6B.

**Author response image 3. sa3fig3:** Western blotting data from additional two pairs of mice in Supplementary Fig.2G.

**Author response image 4. sa3fig4:** Western blotting data from additional two pairs of mice in Supplementary Fig.3A.

**Author response image 5. sa3fig5:** Western blotting data from additional two pairs of mice in Supplementary Fig.3C.

**Author response image 6. sa3fig6:** Western blotting data from additional two pairs of mice in Supplementary Fig.3D.

**Author response image 7. sa3fig7:** Western blotting data from additional two pairs of mice in Supplementary Fig.4F.

**Author response image 8. sa3fig8:** Western blotting data from additional two pairs of mice in.

**Author response image 9. sa3fig9:** Western blotting data from additional two pairs of mice in Supplementary Fig.7C.

Comment 13: Knockout mice should be placed on HFD or keto diet to test for the effects of PTPMT1 depletion.

We appreciate this thoughtful suggestion. We will certainly incorporate this suggestion into our future studies, expanding beyond the scope of the current initial report.

Comment 14: Suggestions on Fig 4A.

Please see our response to Comment 10.

Comment 15: Suggestions for improving echocardiographs.

These analyses were conducted by trained personnel at the Emory Animal Physiology Core. The data presented in our manuscript was provided by them. We appreciate bringing the issues to our attention, and we will inform them accordingly.

Comment 16: Comment on Fig 5B.

The tissues were sectioned at comparable, if not identical, levels. WT and PTPMT1 knockout heart sections look dramatically different because of the dilated myopathy observed in the knockout hearts.

Comment 17: Comment on Fig 5C.

We believe the cell death occurred predominantly in cardiomyocytes.